# A Survey of Token Compression for Efficient Multimodal Large Language Models

**Kele Shao**[*,1,2]                                                     *shaokele@gmail.com*

**Keda Tao**[*,1,2]                                                      *taokeda@westlake.edu.cn*

**Kejia Zhang**[3]                                                       *kejiaz171@gmail.com*

**Sicheng Feng**[2,4]                                                    *fengsicheng@u.nus.edu*

**Mu Cai**[5]                                                            *im.mucai@gmail.com*

**Yuzhang Shang**[6]                                                     *yuzhang.shang@ucf.edu*

**Haoxuan You**[7]                                                       *haoxuanyou@gmail.com*

**Can Qin**[8]                                                           *qin.ca@northeastern.edu*

**Yang Sui**[9]                                                          *yangsui.research@gmail.com*

**Huan Wang**[†,2]                                                       *wanghuan@westlake.edu.cn*

[1] *Zhejiang University*   [2] *Westlake University*   [3] *Xiamen University*   [4] *National University of Singapore*
[5] *University of Wisconsin-Madison*   [6] *University of Central Florida*   [7] *Columbia University*   [8] *Salesforce
AI Research*   [9] *Rice University*   [*] *Equal Contribution*   [†] *Corresponding Author*

**Reviewed on OpenReview:** <https://openreview.net/forum?id=G2od9JVHkE>

## Abstract

Multimodal large language models (MLLMs) have made remarkable strides, largely driven by their ability to process increasingly ***long and complex*** contexts, such as high-resolution images, extended video sequences, and lengthy audio input. While this ability significantly enhances MLLM capabilities, it introduces substantial computational challenges, primarily due to the quadratic complexity of self-attention mechanisms with numerous input tokens. To mitigate these bottlenecks, token compression has emerged as an auspicious and critical approach, efficiently reducing the number of tokens during both training and inference. In this paper, we present the first systematic survey and synthesis of the burgeoning field of multimodal long context token compression. Recognizing that effective compression strategies are deeply tied to the unique characteristics and redundancies of each modality, we categorize existing approaches by their primary data focus, enabling researchers to quickly access and learn methods tailored to their specific area of interest: *(1) image-centric compression*, which addresses spatial redundancy in visual data; *(2) video-centric compression*, which tackles spatio-temporal redundancy in dynamic sequences; and *(3) audio-centric compression*, which handles temporal and spectral redundancy in acoustic signals. Beyond this modality-driven categorization, we further dissect methods based on their underlying mechanisms, including ***transformation-based***, ***similarity-based***, ***attention-based***, and ***query-based*** approaches. By providing a comprehensive and structured overview, this sur-

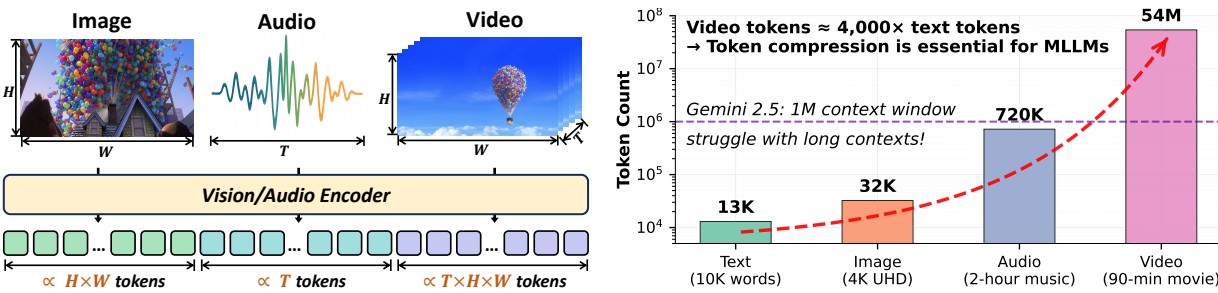

Figure 1: **Left:** Image, video, and audio data types can scale in their representation dimensions, leading to a corresponding increase in the number of tokens. **Right:** Top-performing MLLMs cannot address real-world demands, as the number of tokens for multimodal input, especially video, vastly exceeds that of text, and most visual tokens are redundant. Therefore, token compression is crucial to address this limitation.

vey aims to consolidate current progress, identify key challenges, and inspire future research directions in this rapidly evolving domain.

# 1 Introduction

Multimodal large language models (MLLMs) (Liu et al., 2023; Li et al., 2025a; Xu et al., 2024a; Bai et al., 2023; Xu et al., 2025b; Lin et al., 2024a; Zhang et al., 2023a; Li et al., 2024a; 2023d; Cheng et al., 2024c; Zhang et al., 2025a; Song et al., 2024b) have demonstrated exceptional performance on complex tasks, including visual question answering (VQA), automatic speech recognition (ASR) and multimodal content generation, by extending the architectural of large language models (LLMs) (Chiang et al., 2023; Team, 2024; AI@Meta, 2024; Abdin et al., 2024). These powerful models derive their strength from processing long and diverse contexts, such as high-resolution images, extended video sequences, and long audio input, using transformer architectures.

Achieving this capability, however, faces a significant challenge: the quadratic complexity of the self-attention mechanism. As the number of tokens increases, this complexity leads to substantial computational and memory demands. This problem is particularly pronounced in MLLMs, where the tokenization of visual and audio data can generate sequences of orders of magnitude longer than text (Shao et al., 2025; Tao et al., 2025a; Yang et al., 2025c; Song et al., 2025c).

For instance, as illustrated in Figure 1, the number of image tokens is directly proportional to resolution, while the number of audio tokens is proportional to duration, and video tokens scale with both resolution and duration. A single content-rich video can produce tens of millions of tokens, dramatically exacerbating computational inefficiencies and leading to severe inference latency (90 minutes video will be converted into 54M tokens)[1]. Consequently, addressing this computational bottleneck is critical for unlocking the full potential of MLLMs in real-world applications.

To address the challenges posed by the long context, *token compression* has emerged as a critical research focus for enhancing the inference efficiency and practical deployment of MLLMs. This approach is highly effective because multimodal inputs, like those processed by vision transformers (ViT), contain significant redundancy (Rao et al., 2021; Liang et al., 2022; Bolya et al., 2022; Ryoo et al., 2021; Touvron et al., 2021; Vaswani et al., 2017; Yang et al., 2025d). Extensive research, for example, demonstrates that more than 50% of tokens in a typical MLLM sequence receive minimal attention during inference (Chen et al., 2024a; Huang et al., 2025c; Tao et al., 2025a; Shao et al., 2025; Alvar et al., 2025; Shang et al., 2025). While some advanced techniques integrate compression directly into a model's architecture or training framework (Chen et al., 2024c; Dai et al., 2024; Wang et al., 2024c; Bai et al., 2025; Li et al., 2025a; Zhang et al., 2024d; Cai et al., 2024a; Yao et al., 2024; Cha et al., 2024; Chu et al., 2023; 2024a; Li et al., 2024d), a major advantage of token compression is its ability to be applied as a post-optimization technique without requiring

---

[1] $90\,\mathrm{min} \times 60\,\mathrm{s/min} \times 10\,\mathrm{frames/s} \times 1000\,\mathrm{tokens/frame}$.

expensive retraining. These methods typically operate by first establishing a specialized metric to evaluate token importance, then performing a corresponding pruning or compression. By significantly accelerating inference and reducing memory consumption, these techniques enable the practical deployment of MLLMs in real-world applications (Lin et al., 2025a; Chu et al., 2023; 2024a; Wei et al., 2025; Ma et al., 2024b).

Recent extensive research demonstrates that token compression substantially enhances inference efficiency, driving the continuous development of diverse compression strategies and sophisticated methodologies (Shen et al., 2025a; Chai et al., 2025; Alvar et al., 2025; Huang et al., 2025c; Yang et al., 2025c; Shang et al., 2025; Zhang et al., 2025b; Cao et al., 2023; Yang et al., 2025a; Chen et al., 2024a; Tao et al., 2025c; Zhang et al., 2024c; Liu et al., 2024c; Yang et al., 2025d; Ma et al., 2025b). However, the inherent heterogeneity of multimodal data means that redundancy differs across modalities. Unlike textual prompts, where redundancy is primarily in syntactic or semantic, visual and auditory data exhibit unique structural properties. For instance, high-resolution images contain strong local correlations, while video streams feature extensive spatiotemporal redundancy across frames, and audio signals often contain extended segments of silence or stationary noise. Consequently, most existing methods focus on compressing one or two specific modalities.

Significant strides have been made in compressing tokens in text LLMs. For instance, (Li et al., 2025d) has thoroughly explored prompt compression for text LLMs, highlighting advancements in this domain. In MLLMs, position paper (Kong et al., 2025) has begun to broaden our understanding, emphasizing that token compression offers benefits beyond mere efficiency. Furthermore, some researchers argue that the focus of research for efficient AI is shifting from model-centric compression to data-centric compression (Liu et al., 2025d). However, there has not yet been a systematic classification of token compression methods specifically for MLLMs, leaving an opportunity for a comprehensive survey in this area.

Motivated by the critical need for efficiency in MLLMs and a desire to address this current research fragmentation, this work presents the first comprehensive, structured survey of long-context token compression techniques. We systematically categorize existing approaches according to their primary modality focus:

- **Image-centric** token compression addresses inherent spatial redundancy, leveraging the fact that neighboring patches usually represent similar textures or colors;

- **Video-centric** token compression targets spatiotemporal redundancy, mitigating the significant inter-frame correlation where consecutive frames typically share extensive background elements and limited motion;

- **Audio-centric** token compression mitigates temporal and spectral redundancy, as salient information often concentrates within sparse, brief segments and specific frequency bands amidst silent pauses or background noise.

Importantly, while acknowledging modality-specific influences on redundancy patterns and optimal compression strategies, we observe that fundamental algorithmic principles frequently transcend individual modalities. Effective compression fundamentally centers on *three* core computational objectives: *importance identification*, *redundancy quantification*, and *token merging or pruning*. These objectives manifest similarly across visual, temporal, and auditory domains despite distinct structural constraints. Consequently, we further categorize methodologies according to their underlying mechanisms: transform-based, similarity-based, attention-based, and query-based approaches.

This work presents the first structured survey of token compression techniques for MLLMs, a critical step in navigating their inherent computational complexities. By consolidating current progress, this survey identifies key challenges and illuminates promising future research directions, providing a foundational resource for both researchers and developers.

The remaining sections of the article are organized as follows: we will first discuss the architecture of MLLMs in the background section (Section 2.1), followed by an examination of how token compression has been utilized in prior methods for large language models (LLMs, Section 2.2) and vision transformers (ViTs, Section 2.3). Subsequent sections will be dedicated to token compression methods for specific modalities: Section 3 for image LLMs, Section 4 for video LLMs, and Section 5 for audio LLMs. Following this, Section 6 will provide insights into token compression research. Finally, Section 7 will introduce the broad application space of token compression, followed by the concluding Section 8.

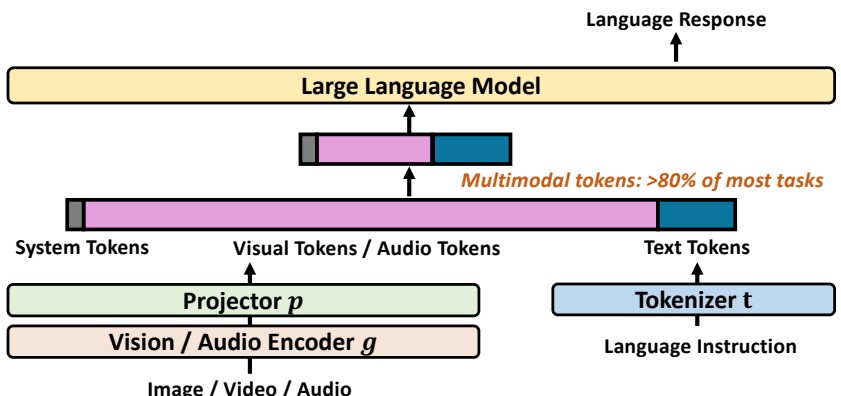

Figure 2: **Representative Architecture of MLLMs.** Within MLLM reasoning processes, token sequences comprise concatenated system tokens, multimodal tokens, and text tokens. Multimodal tokens usually constitute the majority of the sequence tokens.

## 2 Background

### 2.1 Multimodal Architecture

The general multimodal large language model (MLLM) framework (see Figure 2), consists of three core components: (1) a modality-specific encoder ($g$), (2) a projector module ($P$), and (3) a pre-trained large language model (LLM).

The process begins with the modality encoder, $g$, which is responsible for processing a given input, such as a visual or audio signal. This encoder compresses the high-dimensional raw data into a sequence of compact and semantically meaningful patch embeddings. For an input image $X_v$ and an audio $X_a$, this can be expressed as:

$$Z_v = g(X_v), \quad Z_a = g(X_a). \tag{1}$$

The encoding function $g$ is a flexible component that can be specialized for various modalities, including vision, audio, sensor data, etc. Widely adopted encoders implementing this function include:

- **Vision encoders:** CLIP (Radford et al., 2021), SigLIP (Zhai et al., 2023), DINO (Caron et al., 2021; Oquab et al., 2023), and ViT (Bai et al., 2025);

- **Audio encoders:** Whisper (Radford et al., 2023) and Audio-CLIP (Guzhov et al., 2022).

Subsequently, the encoded embeddings ($Z_v$ or $Z_a$) are transformed by the projector module, $P$. The primary role of this module is to bridge the modality gap by mapping the embeddings into the same latent space as the text embeddings of LLM.

$$H_v = P(Z_v), \quad H_a = P(Z_a). \tag{2}$$

The output of the projector, a sequence of projected embeddings, can then be seamlessly concatenated with the text prompts and fed into the LLM.

The pre-trained LLM (Chiang et al., 2023; Team, 2024; AI@Meta, 2024) forms the core of the framework, with its large-scale parameters providing emergent capabilities such as zero-shot generalization and in-context learning. The LLM receives a composite input sequence formed by concatenating the projected multimodal embeddings $H_v$ and $H_a$, as well as the textual prompt embeddings $H_q$. The textual prompt $X_q$ is first converted into embeddings $H_q$ by an integrated tokenizer.

The LLM then generates a response sequence $Y_a$ through autoregressive decoding:

$$p(Y_a|H_v, H_a, H_q) = \prod_{i=1}^{L} p(y_i|H_v, H_a, H_q, y_{<i}), \tag{3}$$

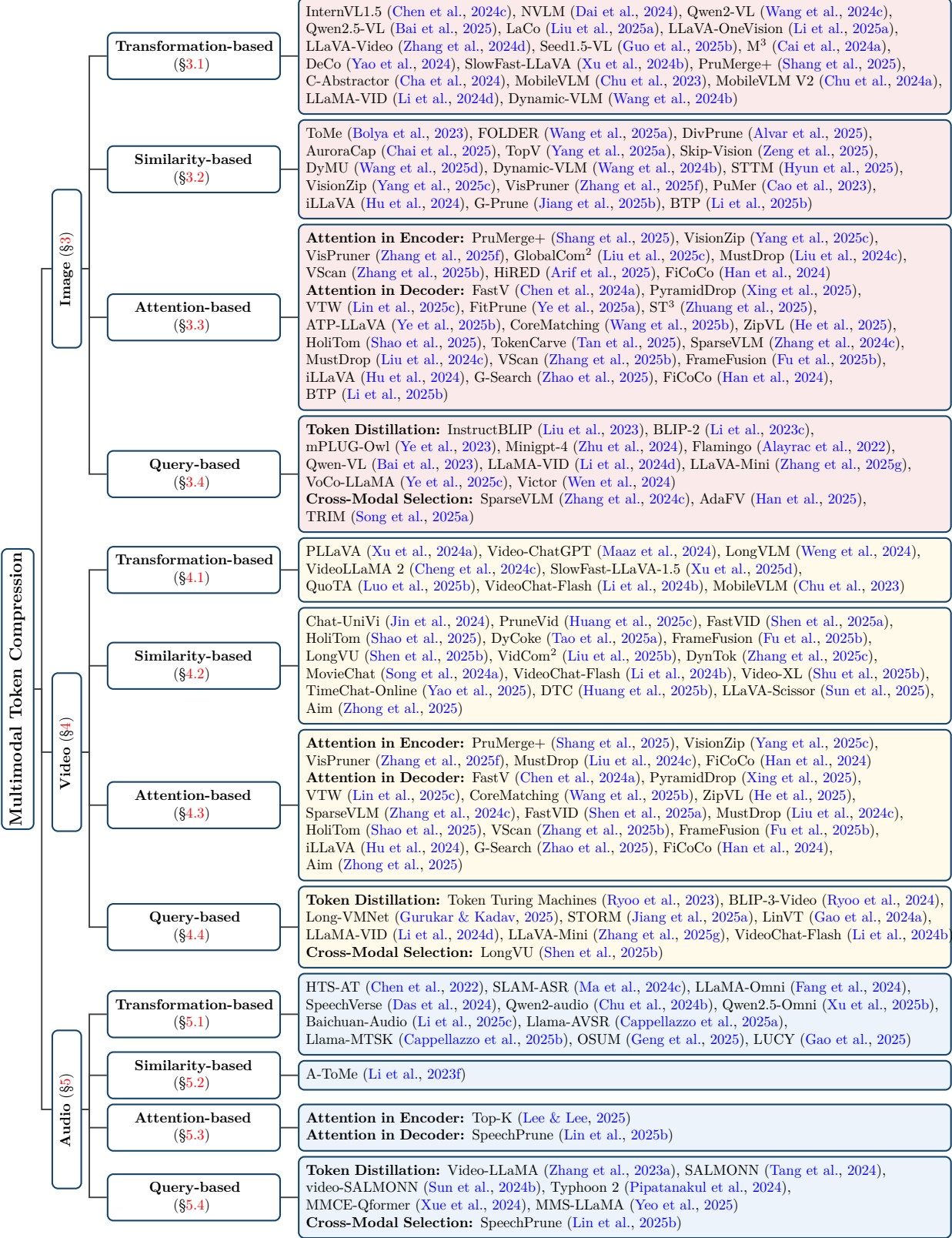

Figure 3: **Taxonomy of Multimodal Token Compression.** Our classification organizes existing methods by their dominant data modality, accounting for inherent differences in redundancy across modalities. This is further refined by a dissection of their underlying mechanisms, enabling researchers to quickly pinpoint methods tailored to specific research domains.

where $L$ signifies the output sequence length.

The high dimensionality of multimodal data poses a computational challenge. As shown in Figure 2, the token sequence processed comprises a mix of system prompt, multimodal context, and textual instruction. In most reasoning tasks, multimodal tokens constitute over 80% of the total sequence length (Chen et al., 2024a), thereby forming the primary computational bottleneck. This bottleneck obstacle to scaling MLLMs and achieving efficient inference. Consequently, a key strategy to optimize computational efficiency involves employing specialized projector architectures. These projectors are designed to reduce the number of multimodal tokens while preserving their semantic fidelity, thus mitigating the computational burden.

While MLLM architecture presents unique challenges, token compression has been explored for both encoders and LLMs independently. Therefore, the subsequent sections will first dive into techniques relevant to these individual components, paving the way for more efficient multimodal models. Specifically, Section 2.2 will focus on token compression methods for large language models (LLMs), and Section 2.3 will explore techniques for vision transformers (ViTs).

## 2.2 Large Language Model Token Compression

The backbone of modern MLLMs is often built upon and fine-tuned from powerful text-based LLMs. As a foundational component, a solid understanding of token compression techniques developed for text LLMs is crucial, as they offer an accurate and lightweight solution for handling real-world long-context scenarios, such as understanding an entire book or a code repository. Within the domain of large language models, these methods are frequently termed prompt compression (Li et al., 2025d).

AutoCompressor (Chevalier et al., 2023) condenses context into summary vectors as soft prompts. Extensible Tokenization (Shao et al., 2024) employs intermediate modules to compress embeddings, while SentenceVAE (An et al., 2024) represents sentences with single tokens. Selective Context (Li et al., 2023g) employs self-information metrics to eliminate low-information tokens. LLMLingua (Jiang et al., 2023a;b; Pan et al., 2024) series utilizes hierarchical token pruning with instruction tuning and further introduces LongLLMLingua (Jiang et al., 2023b) to mitigate position decay through semantic density ranking.

In parallel, query-guided methods like QUITO (Wang et al., 2024e) and QUITO-X (Wang et al., 2024f) leverage attention scores or information bottleneck theory for relevance-based filtering. AdaComp (Zhang et al., 2024a) implements adaptive extraction governed by query complexity predictors. Concept Distillation (Shi et al., 2024) employs Abstract Meaning Representation (AMR) graphs to distill key concepts, whereas xRAG (Cheng et al., 2024b) collapses documents into single-token representations. ICAE (Ge et al., 2023) encodes context into discrete memory slots. Recursive frameworks including RCC (Huang et al., 2024) and XL3M (Wang et al., 2024d) generate piecewise summaries through relevant fusion. SoftPromptComp (Wang et al., 2024a) fuses natural language prompts with dynamic embeddings, while PromptIntern (Zou et al., 2024) internalizes task instructions into model parameters via phased training.

Targeting inference efficiency, KV cache compression techniques prune redundant memory states to accelerate generation. H2O (Zhang et al., 2023b) and StreamingLLM (Xiao et al., 2024) utilize heavy-hitter policies and attention sinks to maintain generation quality under limited budgets. Furthermore, SnapKV (Li et al., 2024e) and PyramidKV (Cai et al., 2024b) enhance long-context performance by pinpointing key attention clusters or dynamically adjusting cache allocations across layers.

While these text-centric token compression techniques have demonstrated notable efficacy, their direct application to MLLMs faces fundamental challenges. The inherent heterogeneity of multimodal data introduces distinct redundancy patterns absent in unimodal text. These include, but are not limited to, spatial correlations in high-resolution images, spatiotemporal continuity in video sequences, and spectral-temporal locality in audio streams. Such specialized redundancies necessitate the development of dedicated compression strategies. Consequently, this survey systematically reviews emerging token compression methodologies for MLLMs that effectively reduce token redundancy while preserving task performance.

### 2.3 Vision Transformer Token Compression

Visual token compression, originally pioneered in vision transformers (ViTs) (Vaswani et al., 2017; Dosovit-skiy et al., 2020; Dong et al., 2022; Liu et al., 2021; Fan et al., 2021; Li et al., 2022; Graham et al., 2021; Huang et al., 2025a; Feng & Zhang, 2023), offers insights for addressing analogous challenges in MLLMs.

Spatial redundancy manifests in ViTs through adjacent image patches, where not all tokens contribute equally to classification outcomes, compounded by semantic imbalance: foreground objects demand disproportionate computational resources compared to homogeneous backgrounds. To mitigate these issues, visual token compression techniques are employed to reduce computational overhead while maintaining model accuracy.

Foundational approaches, including DynamicViT (Rao et al., 2021) and EViT (Liang et al., 2022), quantify token relevance through attention scores, dynamically pruning low-saliency tokens. Complementary techniques like ToMe (Bolya et al., 2022) and TokenLearner (Ryoo et al., 2021) either merge semantically similar tokens using similarity metrics or generate compact token sets via learned spatial attention mechanisms. DeiT (Touvron et al., 2021) employs lightweight 'student' heads to predict categorical labels from compressed token subsets. Furthermore, methods such as MADTP (Cao et al., 2024) leverage cross-modal alignment to filter tokens.

The preceding analysis demonstrates that ViT token compression methodologies offer substantive inspiration for token reduction in MLLMs. However, MLLMs possess not only multimodal tokens encoding low-level features but also text tokens conveying high-level abstractions, coupled with significantly longer token sequences. Consequently, token compression in MLLMs presents greater challenges than in ViT while being increasingly critical for computational efficiency. Therefore, this survey analyzes the evolution and future directions of token compression techniques for MLLMs operating in long-context multimodal environments.

### 2.4 Problem Definition and Taxonomy Scope

To clarify the scope of this survey and distinguish token compression from related efficient computing techniques, we establish a strict criterion based on the physical reduction of information flow. We define a method as token compression if and only if it *explicitly reduces the number of tokens passed to subsequent layers or modules.*

Formally, given an input sequence $\mathbf{X} \in \mathbb{R}^{N \times D}$, a token compression operator $\mathcal{T}$ produces an output $\mathbf{X}' \in \mathbb{R}^{M \times D}$ where $M < N$, while aiming to retain the essential semantic information of $\mathbf{X}$. Based on this definition, we delineate the boundaries with related concepts as follows:

- Input-level Compression: We classify techniques such as frame sampling and key-frame extraction as a generalized form of token compression operating at the Input Level. By selecting a subset of frames (e.g., extracting key-frames from a video), the initial token count $N$ is reduced prior to the encoding stage. We distinguish this from Feature-level Compression (e.g., token pruning or merging), which dynamically operates on intermediate embeddings within the network layers.

- Exclusion of Attention Sparsity: We exclude attention sparsity mechanisms (and related efficient attention variants) from the scope of token compression. While these methods reduce computational complexity (e.g., from $O(N^2)$ to linear) by masking interactions, they typically output a sequence of the same length ($N \to N$) to the next layer. They sparsify the computation graph, whereas token compression sparsifies the representation.

## 3 Image-centric Token Compression

Multimodal long context token compression methods generally fall into *four* categories based on their underlying mechanisms: **transformation-based** approaches directly transform the cross-modality information to compress tokens by altering their scale or representation; **similarity-based** techniques reduce tokens

Table 1: Four Categories of Methods Based on Intrinsic Mechanisms: Diagram, Summary, and Pros & Cons.

| Method | Transformation-based | Similarity-based | Attention-based | Query-based |
|---|---|---|---|---|
| Diagram |  e.g., Pool, Conv |  e.g., KNN |  Attention Matrix |  |
| Summary | Transform tokens into a more compact form | Compress by merging or grouping similar tokens | Remove less attentive tokens via attention sparsity | Use external queries to guide token compression |
| Pros | Preserve the structural representation of information well | Simplify processing, flexibility in choosing where to compress | Dynamically prune tokens by relevance; tie to original computation, boosting interpretability | Suitable for specific and video tasks, as compressed information is more relevant and concise |
| Cons | Limited by the transformation method, compression rate isn't flexible enough | May lose fine-grained info if tokens overgeneralized; poor structural feature retention | Explicit attention score calculation might be incompatible with mainstream acceleration libraries | Not user-friendly for multi-turn conversations; requires recondensing information |

by leveraging the inherent resemblances between them; **attention-based** strategies exploit the sparsity of attention within the multimodal data to guide compression; and **query-based** methods selectively refine multimodal information, guided by prompts, to distill the most relevant tokens. Each of these methods has its own set of advantages and disadvantages, which are summarized in Table 1. Representative image-centric token compression methods are further compared in Table 2.

## 3.1 Transformation-based Image-centric Compression

Transformation-based image-centric compression methods leverage the spatial redundancy inherent in 2D image representations. Some image token compression techniques are derived from image downsampling operations (*e.g.*, pooling, bilinear interpolation). Based on the specific transformation method, these can be broadly categorized as follows:

### 3.1.1 Pixel Unshuffle

Pixel unshuffle is the inverse operation of pixel shuffle. It transforms a feature map from a high spatial resolution with a small number of channels into a lower-resolution feature map with a larger number of channels. This reduces the number of tokens. The transformation can be mathematically expressed as:

$$\text{Pixel Unshuffle: } H \times W \times D \rightarrow \frac{H}{r} \times \frac{W}{r} \times (D \cdot r^2), \tag{4}$$

where $H$, $W$ denote the height and width of the token grid, $D$ is the hidden dimension of each token, and $r$ is the downsampling ratio. Here $r$ is a positive integer, typically 2. Therefore, as summarized in Table 1, the token compression ratio for transformation-based methods is usually limited to a few specific values, generally compressing the number of tokens to 25%.

Recent works like InternVL series (Chen et al., 2024c;b; Gao et al., 2024b; Zhu et al., 2025a), Qwen2 series (Wang et al., 2024c; Bai et al., 2025), and NVLM (Dai et al., 2024) utilize pixel unshuffle to reduce the tokens generated by the vision tower by a factor of one-quarter. Subsequently, an MLP is employed to align the visual dimension with the text dimension, addressing the mismatch in the hidden dimension.

### 3.1.2 Spatial Pooling / Interpolation

Unlike pixel unshuffle, pooling and interpolation directly perform 2D downsampling on tokens, without altering the hidden dimension. This process can be defined as:

$$\text{Pooling / Interpolation: } H \times W \times D \rightarrow \frac{H}{S} \times \frac{W}{S} \times D, \tag{5}$$

where $S$ is the downsampling factor.

LLaVA-OneVision (Li et al., 2025a) employs bilinear interpolation for 2D downsampling of aligned tokens, while LLaVA-Video (Zhang et al., 2024d) uses average pooling for downsampling. $M^3$ (Cai et al., 2024a) utilizes a simple pooling operation to learn an inherently multi-granular representation during training. This allows the model to achieve comparable performance with fewer tokens during inference, effectively addressing efficiency concerns. DeCo (Yao et al., 2024) argues that the Q-former (Liu et al., 2023; Li et al., 2023c) is an inefficient visual compressor and similarly achieves token compression through a straightforward average pooling approach, leading to improved convergence efficiency and performance.

### 3.1.3 Spatial Convolution

Convolutional operations offer a more sophisticated approach to token compression compared to simple pooling or interpolation, by learning to abstract local information while reducing spatial dimensions. The transformation can be expressed as:

$$\text{Convolution: } H \times W \times D_{in} \rightarrow \frac{H}{S} \times \frac{W}{S} \times D_{out}, \tag{6}$$

where S is the stride, which determines the downsampling factor, and $D_{in}$, $D_{out}$ represent the input and output channel dimensions, respectively.

Honeybee (Cha et al., 2024) proposes the C-Abstractor, which uses convolution to extract and compress token information while preserving locality. MobileVLM (Chu et al., 2023), on the other hand, employs an LDP module that utilizes depth-wise convolution to reduce the number of tokens by 75%.

### 3.1.4 Comparative Analysis of Transformation Methods

These transformation-based image-centric compression methods effectively utilize all image tokens while consciously preserving the spatial local information of 2D features. Pixel unshuffle, pooling, and interpolation are inherently parameter-free, thus introducing no additional weight overhead, a key advantage. In contrast, convolution learn a more sophisticated local abstraction by introducing trainable weights.

Another notable difference lies in how these methods handle feature dimensions: pixel unshuffle typically alters the hidden dimension, necessitating a subsequent trained MLP to align with the text dimension. Conversely, pooling and interpolation can be implemented in a training-free manner as they operate directly on the aligned token dimension.

By extracting more condensed information, they achieve a superior balance between performance and efficiency. However, due to the inherent characteristics of 2D downsampling, their token compression ratios are typically limited to a few specific magnitudes, with a 25% compression rate being the most common.

### 3.2 Similarity-based Image-centric Compression

Similarity-based image-centric compression methods reduce the number of visual tokens by identifying and merging similar tokens based on their distance or similarity in an implicit space. This typically involves selecting representative cluster-center tokens to encapsulate visual information.

Early works in this area include ToMe (Bolya et al., 2023), an acceleration method for ViTs. ToMe introduces a token merge module between the attention and MLP blocks, calculating token similarity and merging similar tokens via bipartite soft matching. This process creates a new set of tokens, $\mathcal{T}$, by replacing the most

similar tokens with their merged representations.

$$\mathcal{T} = (\mathcal{T}_{original} \setminus \bigcup_{i=1}^{k} C_i) \cup \{\text{Merge}(C_i)\}_{i=1}^{k}, \tag{7}$$

where each $C_i$ is a set of tokens identified as highly similar by ToMe's matching algorithm.

In the context of MLLMs, FOLDER (Wang et al., 2025a) employs a similar approach, inserting a token merge module within the last attention block of the vision encoder. This reduces the number of tokens that were subsequently passed to the LLM decoder. DivPrune (Alvar et al., 2025) reframes the token compression problem as a Max-Min diversity problem (Porumbel et al., 2011), aiming to select a subset of tokens with maximal internal differences. AuroraCap (Chai et al., 2025) adopts a strategy consistent with ToMe, performing token merging within each attention and MLP block of the vision tower. This progressively reduces the number of tokens throughout the ViT model. While the aforementioned methods primarily leverage similarity-based clustering of tokens within the ViT, TopV (Yang et al., 2025a) extends this principle to compress tokens within the LLM layers. TopV comprehensively considers both the similarity and distance functions between features to guide the token compression process, operating directly within the multimodal representation space of the LLM.

### 3.2.1 Analysis of Similarity Methods

While similarity-based methods effectively reduce tokens, this merging often overlooks the original spatial information of the tokens, leading to spatial misunderstanding (in Tab. 1). Subsequent work frequently employs methods like DPC-KNN (Du et al., 2016; Rodriguez & Laio, 2014) or techniques focused on local spatial similarity merging to prevent excessive spatial information degradation. Furthermore, when tokens are over-generalized, similarity-based methods struggle to distinguish between them, easily leading to misjudgment.

### 3.3 Attention-based Image-centric Compression

Attention-based token compression methods leverage the inherent sparsity of visual feature attention to guide token pruning. Tokens with low attention scores can often be considered removable without significantly impacting the original computation. Specifically, these methods utilize the attention mechanism to identify and preserve pivotal tokens. It is worth noting that this shares the same underlying philosophy as sparse attention methods (Yuan et al., 2025; Zhang et al., 2025d; Lu et al., 2025; Yin et al., 2025), which focus on executing the critical attention computations, yet manifest at a different scale: the former operates on token quantity while the latter operates on computational pathways. In vision language models, both the vision encoder and the LLM decoder incorporate transformers. Consequently, attention-based compression strategies can be broadly categorized into those applied within the encoder and those within the decoder.

### 3.3.1 Attention in Encoder

Methods focusing on the vision encoder primarily select visual tokens based on attention scores within a single image or crops, relying on the capabilities of the vision transformer (ViT). This reduces the number of visual tokens before they're passed to the LLM. To achieve this, the set of retained tokens, $\mathcal{T}_{\text{encoder}}$, is determined by selecting the top $k$ tokens based on their attention scores relative to the [CLS] token:

$$\mathcal{T}_{\text{encoder}} = \text{TopK}_k\left(\{\text{Attention}\left(\mathbf{v}_i, \mathbf{v}_{\text{cls}}\right) \mid \mathbf{v}_i \in \mathcal{V}\}\right), \tag{8}$$

where $\mathcal{V}$ is the original set of visual tokens, $\mathbf{v}_i$ is the $i$-th visual token, and $\mathbf{v}_{\text{cls}}$ is the [CLS] token. This strategy ensures that only the most salient visual information, as highlighted by the [CLS] attention, is carried forward for further processing.

Prumerge (Shang et al., 2025) selects cluster centers for visual tokens based on [CLS] attention in the encoder. It then merges the remaining less attentive tokens using K-nearest neighbors (KNN) clustering and a weighted cluster center update mechanism. VisionZip (Yang et al., 2025c) retains visual tokens with high attention

Table 2: Comparison of Training-Free Token Compression Methods for Image LLMs in Understanding Tasks

| Method | #Vision Tokens | Res. | VQA$^2$ | GQA | VisWiz | SciQA | VQA$^T$ | POPE | MME | MMB | SEED | LLaVA$^W$ | MM–Vet |
|---|---|---|---|---|---|---|---|---|---|---|---|---|---|
| BLIP-2 (Li et al., 2023c) | 32 | 224 | 65.0 | 41.0 | 19.6 | 61.0 | 42.5 | 85.3 | 1293.8 | – | 46.4 | 38.1 | 22.4 |
| IDEFICS-9B (Laurençon et al., 2023) | 64 | 224 | 50.9 | 38.4 | 35.5 | – | 25.9 | – | – | 48.2 | – | – | – |
| MobileVLM-3B (Chu et al., 2024a) | 144 | 336 | – | 59.0 | – | 61.0 | 47.5 | 84.9 | 1288.9 | 59.6 | – | – | – |
| mPLUG-Owl2 (Ye et al., 2023) | 1024 | 448 | 79.4 | 56.1 | 54.5 | 68.7 | 54.3 | – | 1450.2 | 64.5 | 57.8 | – | 36.2 |
| Video-LLaVA (Lin et al., 2024a) | 256 | 224 | 74.7 | 60.3 | 48.1 | 66.4 | 51.8 | 84.4 | – | 60.9 | – | 73.1 | 32.0 |
| Qwen-VL (Wang et al., 2024c) | 256 | 448 | 78.8 | 59.3 | 35.2 | 67.1 | 63.8 | – | – | 38.2 | 56.3 | – | – |
| LLaVA-v1.5 (Liu et al., 2023) | 576 | 336 | 78.5 | 62.0 | 50.0 | 66.8 | 58.2 | 85.9 | 1510.7 | 64.3 | 58.6 | 63.4 | 30.5 |
| *LLaVA-v1.5-7B w/ Token Compression Methods (Training Free)* | | | | | | | | | | | | | |
| ToMe (Bolya et al., 2023) | 192 | 336 | 68.0 | 54.3 | – | – | 52.1 | – | 1563.0 | 60.5 | – | – | – |
| FastV (Chen et al., 2024a) | 192 | 336 | 67.1 | 52.7 | – | – | 52.5 | 64.8 | 1612.0 | 61.2 | 57.1 | – | 27.7 |
| SparseVLM (Zhang et al., 2024c) | 192 | 336 | 75.6 | 57.6 | – | – | 56.1 | 83.6 | 1721.0 | 62.5 | 55.8 | – | 31.5 |
| MustDrop (Liu et al., 2024c) | 192 | 336 | 76.0 | 58.2 | 51.4 | – | 56.5 | – | 1787.0 | 62.3 | – | – | – |
| PruMerge+ (Shang et al., 2025) | 144 | 336 | 76.8 | – | – | 68.3 | 57.1 | 84.0 | 1462.4 | 64.9 | – | – | – |
| ATP-LLaVA (Ye et al., 2025b) | 144 | 336 | 76.4 | 59.5 | – | – | – | 84.2 | 1473.9 | 66.0 | 57.3 | – | 31.5 |
| VisionZip++ (Yang et al., 2025c) | 128 | 336 | 76.6 | 58.9 | – | – | 57.0 | 83.7 | 1823.0 | – | 55.8 | – | 32.9 |
| VisPruner (Zhang et al., 2025f) | 128 | 336 | 75.8 | 58.2 | 52.7 | – | 57.0 | 84.6 | 1461.4 | 62.7 | – | – | 33.7 |
| VisionZip++ (Yang et al., 2025c) | 64 | 336 | 74.2 | 57.0 | – | – | 56.0 | 80.9 | 1756.0 | – | 53.4 | – | 30.2 |
| TokenCarve (Tan et al., 2025) | 64 | 336 | 74.8 | – | – | – | 57.0 | 79.9 | 1754.0 | 62.0 | – | – | 29.3 |
| VisPruner (Zhang et al., 2025f) | 32 | 336 | 67.7 | 52.2 | 53.0 | – | 53.9 | 72.7 | 1271.0 | 58.4 | – | – | 28.8 |
| *MLLMs w/ Token Compression* | | | | | | | | | | | | | |
| LLaMA-VID (Li et al., 2024d) | 2 | 336 | – | 55.5 | – | 68.8 | 49.0 | 83.1 | – | – | – | – | – |
| LLaVA-Mini (Zhang et al., 2025g) | 1 | 336 | 77.6 | 60.9 | 56.2 | 70.4 | 57.0 | 84.7 | 1466.0 | 65.6 | 58.5 | 68.9 | 36.6 |
| VoCo-LLAMA (Ye et al., 2025c) | 1 | 336 | 72.3 | 57.0 | – | 65.4 | – | 81.4 | 1323.3 | 58.8 | 53.7 | – | – |

scores and subsequently merges the remaining tokens through clustering. VisPruner (Zhang et al., 2025f) similarly preserves a subset of high-attention visual tokens. Then it progressively removes duplicates based on similarity in multiple rounds, ultimately retaining an additional set of diverse tokens. GlobalCom$^2$ (Liu et al., 2025c) employs a hierarchical strategy. It coordinates the attention scores of thumbnail tokens to guide the pruning of high-resolution crops, thereby achieving effective global context reduction.

### 3.3.2 Attention in Decoder

Unlike attention-based compression within the encoder, methods focusing on attention in the decoder leverage the capabilities of the LLMs to guide token compression. Here, attention is computed across all tokens within the LLM's attention window, which includes not only visual tokens but also textual tokens. This allows the LLM to determine the importance of visual and textual information in a joint space, leading to more context-aware token pruning.

A common approach for compression in the decoder involves selecting the most salient visual tokens. The set of retained tokens, $\mathcal{T}_{\text{decoder}}$, is typically determined by choosing the top $k$ visual tokens based on the average attention they receive from all other tokens in that layer's attention window:

$$\bar{A}(\mathbf{v}_i) = \frac{1}{|\mathcal{S}|} \sum_{\mathbf{s}_j \in \mathcal{S}} \text{Attention}\left(\mathbf{v}_i, \mathbf{s}_j\right), \tag{9}$$

$$\mathcal{T}_{\text{decoder}} = \text{TopK}_k\left(\{\bar{A}(\mathbf{v}_i) \mid \mathbf{v}_i \in \mathcal{V}\}\right), \tag{10}$$

where $\mathcal{V}$ denotes the set of visual tokens, and $\mathcal{S}$ represents the entire set of tokens present in the current layer's attention window (which may include visual, textual, or special tokens). This method allows the model to prioritize the visual information that is most relevant in the ongoing context.

FastV (Chen et al., 2024a) is among the first to identify a significant inefficiency in large vision language models (LVLMs), namely the extremely low attention efficiency of visual tokens. For instance, in LLaVA-v1.5, visual tokens received only 0.21% of the attention obtained by system prompts after the second layer. FastV posits that this is due to an overabundance of visual signals, leading to specific features aggregating onto "anchor" tokens via shallow self-attention mechanisms. Consequently, pruning 50% of visual tokens based on attention scores after the second layer maintains maximal performance. PyramidDrop (Xing et al., 2025) structures the token compression process within the LLM into multiple stages. It employs progressive token compression to avoid excessive loss of visual information in shallower layers. VTW (Lin et al., 2025c)

takes a more aggressive pruning approach, arguing that visual tokens can be entirely removed after a certain layer within the LLM. The specific layer for visual token removal is determined using a calibration dataset. FitPrune (Ye et al., 2025a) focuses on reducing the length of visual tokens per layer. It considers both the self-attention of visual tokens and their cross-attention with text tokens to guide compression. The goal is to find an optimal pruning "recipe" that minimizes the distributional gap before and after pruning. $ST^3$ (Zhuang et al., 2025) dynamically reduces tokens during the generation process. It also progressively prunes inattentive visual tokens as the layer goes deeper. ATP-LLaVA (Ye et al., 2025b) introduces an adaptive token pruning (ATP) module within the decoder layers. This module trains threshold heads to adaptively predict pruning thresholds for the current layer and instance, thereby removing redundant or text-irrelevant visual tokens. ZipVL (He et al., 2025) achieves progressive compression by determining the compression ratio for each layer based on its attention score distribution. This allows for a granular and adaptive reduction of visual tokens throughout the model.

### 3.3.3 Critical Challenge for Pruning in Decoder

While these methods leverage attention scores within the LLM decoder to offer sophisticated ways to compress visual tokens, they face a significant practical challenge: the explicit need to access attention scores. This direct access is often incompatible with highly optimized acceleration libraries like FlashAttention (Dao et al., 2022; Dao, 2024), which compute attention implicitly or in a fused manner for speed. This incompatibility can be mitigated by performing an additional, separate attention calculation solely for pruning purposes. However, for progressive pruning strategies such as FitPrune, ST3, and ZipVL, this additional computational overhead becomes significantly more pronounced, potentially negating the efficiency gains.

### 3.4 Query-based Image-centric Compression

Visual information often contains a substantial amount of features irrelevant to the given query. Query-based image-centric compression leverages the query prompt to guide the compression of visual tokens. These methods can be broadly categorized into two types: (1) **Token Distillation:** These methods compress visual tokens by distilling visual tokens into a specific, reduced number of tokens. (2) **Cross-Modal Selection:** These approaches compress tokens by matching between modality-aligned visual and text tokens.

### 3.4.1 Token Distillation

Token distillation originates from the early projector designs of MLLM. The goal is to distill visual tokens to learn the most text-relevant visual representations, reduce visual tokens while also aligning modalities.

The Q-Former series (Liu et al., 2023; Li et al., 2023c), a pioneering approach, uses learnable queries and cross-attention to extract pertinent visual cues from visual features. Similarly, mPLUG-Owl (Ye et al., 2023), MiniGPT-4 (Zhu et al., 2024), Flamingo (Alayrac et al., 2022), and Qwen-VL (Bai et al., 2023) all employ variations of learnable query-based architectures to condense visual information into a smaller fixed set of tokens that are then aligned with the language model. LLaMA-VID (Li et al., 2024d) employs a highly aggressive approach to visual token compression. For a single image or video frame, it utilizes context attention where the text query aggregates text-related visual cues from the visual embedding. Ultimately, it represents an entire image's information using only two tokens. LLaVA-Mini (Zhang et al., 2025g) achieves comparable performance by pre-fusing visual information directly into text tokens, requiring just one visual token. While previous methods relied on external modules for visual token compression, VoCo-LLaMA (Ye et al., 2025c) is notable as the first approach to use LLMs themselves for visual token compression. It distills the LLM's understanding of visual tokens into the processing of VoCo tokens via attention distillation. Victor (Wen et al., 2024) introduces a small number of learnable "register tokens" after the visual tokens. It then uses the shallow layers of a large model to distill visual information into these registers, discarding all original visual tokens to significantly improve inference and training efficiency.

### 3.4.2 Cross-Modal Selection

Cross-modal selection aims to reduce the number of tokens in one modality by leveraging aligned tokens from another. This compression is achieved by identifying and retaining only the most relevant information

Table 3: Comparison of Training-Free Token Compression Methods for Video LLMs in Understanding Tasks

| Method | #Token Ratio | ActivityNet | | Video-ChatGPT | | | | | Next-QA | EgoSchema | LongVideoBench | VideoMME | MVBench |
|---|---|---|---|---|---|---|---|---|---|---|---|---|---|
| | | Acc. | Score | CI | DO | CU | TU | CO | mc | ↑ | ↑ | ↑ | ↑ |
| LLaVA-OneVision (Li et al., 2025a) | 100% | 48.09 | 3.47 | 3.37 | 3.78 | 3.52 | 3.02 | 2.63 | 81.33 | 60.4 | 56.4 | 58.6 | 58.3 |
| *50% Visual Token Retained Ratio* | | | | | | | | | | | | | |
| FastV (Chen et al., 2024a) | 50% | 47.95 | 3.47 | 3.36 | 3.77 | 3.50 | 2.99 | 2.57 | 81.1 | 58.0 | – | 57.5 | – |
| DyCoke (Tao et al., 2025a) | 50% | 47.88 | 3.47 | 3.33 | 3.76 | 3.51 | 3.01 | 2.58 | 81.1 | 57.7 | – | 57.4 | 57.5 |
| PLLaVA (Xu et al., 2024a) | 50% | 47.59 | 3.45 | 3.36 | 3.73 | 3.52 | 3.00 | 2.66 | 81.0 | 57.7 | – | 56.9 | – |
| LLaVA-Scissor (Sun et al., 2025) | 50% | 47.89 | 3.47 | 3.37 | 3.76 | 3.47 | 3.00 | 2.65 | 81.12 | 57.6 | – | 57.4 | – |
| *25% – 35% Visual Token Retained Ratio* | | | | | | | | | | | | | |
| FastV (Chen et al., 2024a) | 35% | 47.83 | 3.46 | 3.32 | 3.74 | 3.47 | 2.97 | 2.61 | 80.5 | 57.8 | – | 56.0 | 61.3 |
| DyCoke (Tao et al., 2025a) | 35% | 47.81 | 3.45 | 3.31 | 3.74 | 3.46 | 2.98 | 2.54 | 80.9 | 57.7 | – | 56.2 | 61.8 |
| PLLaVA (Xu et al., 2024a) | 35% | 47.23 | 3.42 | 3.26 | 3.70 | 3.39 | 2.92 | 2.59 | 79.66 | 56.07 | – | 54.26 | 59.5 |
| VisionZip (Yang et al., 2025c) | 25% | – | – | – | – | – | – | – | – | 60.3 | 56.5 | 58.2 | 57.9 |
| PruneVid (Huang et al., 2025c) | 25% | – | – | – | – | – | – | – | – | 59.9 | 55.7 | 57.4 | 57.4 |
| FastVID (Shen et al., 2025a) | 25% | – | – | – | – | – | – | – | – | – | 56.3 | 58.0 | 56.5 |
| LLaVA-Scissor (Sun et al., 2025) | 25% | 47.79 | 3.47 | 3.33 | 3.76 | 3.47 | 2.98 | 2.62 | 80.66 | 57.64 | – | 56.44 | – |
| HoliTom (Shao et al., 2025) | 25% | – | – | – | – | – | – | – | – | 61.2 | 56.7 | 58.9 | 58.4 |
| *5% – 15% Visual Token Retained Ratio* | | | | | | | | | | | | | |
| VisionZip (Yang et al., 2025c) | 15% | – | – | – | – | – | – | – | – | 59.8 | 54.4 | 56.1 | 56.5 |
| PruneVid (Huang et al., 2025c) | 10% | – | – | – | – | – | – | – | – | 59.8 | 54.5 | 56.0 | 56.2 |
| FastVID (Shen et al., 2025a) | 10% | – | – | – | – | – | – | – | – | – | 56.3 | 57.3 | 55.9 |
| LLaVA-Scissor (Sun et al., 2025) | 10% | 47.75 | 3.46 | 3.26 | 3.68 | 3.41 | 2.90 | 2.52 | 80.0 | 57.5 | – | 55.2 | 57.9 |
| HoliTom (Shao et al., 2025) | 10% | – | – | – | – | – | – | – | – | 61.2 | 56.3 | 56.8 | 57.3 |

across modalities, leading to more efficient and effective processing. Several notable approaches have been proposed to address this challenge:

SparseVLM (Zhang et al., 2024c) employs visual tokens to pre-select relevant text tokens. By leveraging the visual modality as an initial filter, SparseVLM efficiently narrows down the textual search space, focusing on information pertinent to the visual content. AdaFV (Han et al., 2025) employs a dual-metric approach for selecting the most informative visual tokens. It calculates both text-to-image similarity and visual saliency extracted from the vision encoder. By combining these two indicators, AdaFV identifies visual tokens that are not only semantically aligned with the text but also visually prominent or significant. TRIM (Song et al., 2025a) introduces a unique method that begins by identifying outlier tokens based on the similarity between text and visual tokens; these outliers are deemed important. Subsequently, a clustering algorithm is utilized to merge the remaining, less critical tokens. This approach prioritizes distinct, highly relevant tokens before consolidating the rest.

### 3.4.3 Analysis of Similarity Methods

While query-based methods can precisely retain query-relevant tokens compared to the three prior approaches, they are unsuitable for multi-turn QA scenarios. This is because the initial query's token retention is based on its specific question. A subsequent query may target different tokens, necessitating a re-execution of the token compression process. This makes the approach highly inefficient for multi-turn conversations.

## 4 Video-centric Token Compression

Processing long high-definition (HD) videos poses significant challenges for VLMs due to the immense number of tokens generated, far exceeding those from high-resolution images. Unlike image-centric compression, video inherently possesses an additional temporal redundancy. While capturing complete temporal information typically requires a frame rate of at least 24 frames per second (FPS), processing a 10-minute HD video at even 1 FPS still yields token sequences orders of magnitude larger than those from high-resolution images, rendering conventional transformer-based MLLMs impractical for real-world deployment over the videos.

To address this, current video LLMs commonly employ a 1 FPS sampling rate to reduce token counts. Furthermore, unlike methods for single images, which often encode both the global image and a series of local patches for detailed feature extraction, video processing often foregoes this detailed frame-level

segmentation to keep token numbers manageable. Even with these strategies, the quantity of video tokens remains substantial. During model training and understanding, **transformation-based** methods, such as the pooling technique used in LLaVA-Video (Zhang et al., 2024d), are usually employed to reduce tokens and aid the model's comprehension of video content.

Beyond training-time optimizations, alternative approaches primarily focus on post-training optimization. Specifically, **similarity-based** and **attention-based** methods offer generic compression techniques for pre-trained video MLLMs. These methods process encoded token sequences without modifying model weights, enabling plug-and-play acceleration across diverse architectures. By dynamically identifying critical spatio-temporal regions and pruning redundant tokens, these techniques significantly enhance the practicality of video MLLMs for real-world applications.

To fully grasp token compression for video LLMs, it is recommended to first review Section 3, which details spatial compression methods. Next, we will primarily discuss techniques addressing the temporal domain. Similar to image-centric methods, selected video-centric token compression methods are compared in Table 3.

### 4.1 Transformation-based Video-centric Compression

Like image LLMs, video LLMs use encoders for visual tokens. Consequently, transformation-based video-centric compression methods fundamentally operate on the principles established in Section 3.1, with the added capability of performing 3D transformations. A multitude of models showcase cross-modal applicability, performing effectively in both image and video inference tasks. Following the structure of Section 3.1, we will now detail transformation-based video-centric compression methods.

#### 4.1.1 2D/3D Pooling

In video LLMs, token pooling is a crucial strategy for managing the high dimensionality of video data. While 2D spatial pooling, as seen in LLaVA-Video (Zhang et al., 2024d), can effectively reduce the token count within individual frames, its efficacy alone may be limited for long-duration videos. A growing number of video LLMs, including PLLaVA (Xu et al., 2024a), Video-ChatGPT (Maaz et al., 2024), SlowFast-LLaVA (Xu et al., 2025d), and LongVLM (Weng et al., 2024), consequently emphasize temporal pooling, which involves downsampling at the frame level.

Notably, PLLaVA demonstrates that model performance exhibits greater sensitivity to temporal pooling than to spatial pooling, highlighting its critical role. For extremely long video sequences, LLaMA-VID (Li et al., 2024d) employs a more aggressive adaptive pooling approach. This method intelligently maintains original resolution for single-image inputs but compresses each video frame to a single token during extended sequence processing, achieving substantial data reduction while aiming to preserve essential information.

This dual focus on spatial and increasingly on temporal pooling underscores their combined importance in enabling efficient processing and comprehensive understanding of video content, particularly as video durations extend. SlowFast-LLaVA (Xu et al., 2025d) incorporates a two-stream SlowFast projector into a LLaVA-style architecture, using a slow pathway to sample fewer, spatially rich frames and a fast pathway to sample more, spatially compressed frames, then concatenates both for the LLM—achieving efficient long-form video understanding with reduced token count while preserving spatiotemporal details.

#### 4.1.2 2D/3D Convolution

Similar to pooling, convolution can also be employed for downsampling video tokens, but it does so in a parameterized manner. Instead of simply aggregating information like pooling, convolution layers learn filters to process and condense spatial and temporal features. VideoLLaMA 2 (Cheng et al., 2024c), for instance, thoroughly investigated both 2D and 3D pooling and convolution approaches. Their experiments showed that 3D convolution yielded the best balance of performance and efficiency for video token downsampling. This suggests that learning intricate spatiotemporal relationships through convolutions is more effective for comprehensive video understanding compared to pooling alone.

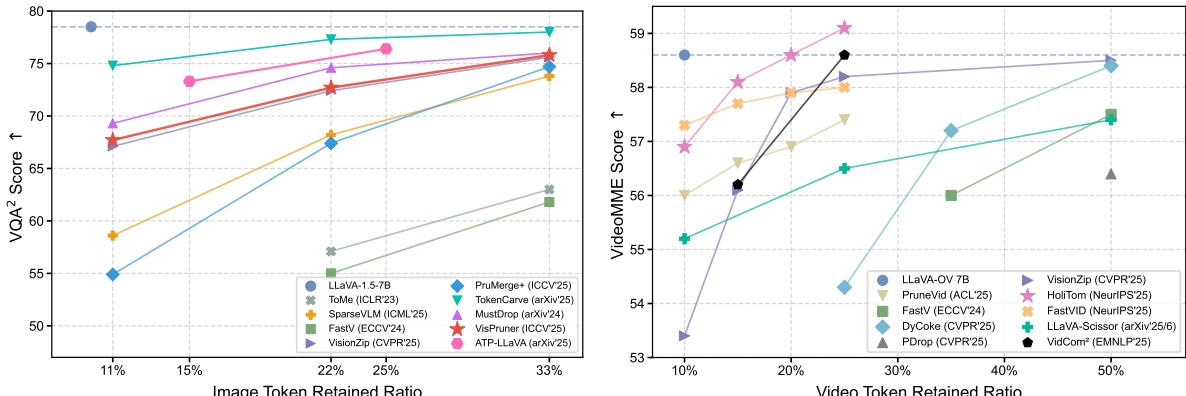

Figure 4: **Trade-off between Retained Ratio and Performance across Modalities.** *Left:* We visualize changes in token retention and model performance on the VQA$^2$ (Goyal et al., 2017) for image LLMs using each method's reported setup with *LLaVA-1.5-7B* (Liu et al., 2023). *Right:* For video LLMs, we plot the video-token retention ratio and the corresponding performance deltas on the VideoMME benchmark (Fu et al., 2025a), following each method's reported configuration with *LLaVA-OV-7B* (Li et al., 2025a). As different methods target distinct compression regimes, we primarily report results at the compression rates specified in their original papers.

## 4.2 Similarity-based Video-centric Compression

Given the temporal redundancy inherent in video, where adjacent frames often exhibit high visual similarity, temporal compression is frequently prioritized over or integrated with spatial compression. To effectively handle this temporal redundancy, video frames are typically first clustered.

Chat-UniVi (Jin et al., 2024) initially pools each video frame into a single frame-level representation token. It then utilizes DPC-KNN (Du et al., 2016; Rodriguez & Laio, 2014) (density peak clustering based on K-nearest neighbors) to amalgamate non-essential frames based on these frame representation tokens. Within each resulting cluster, tokens from multiple frames are further clustered to obtain concise spatiotemporal visual representations. Similarly, FastVID (Shen et al., 2025a) divides video frames solely based on the similarity of their adjacent frame representation tokens. It then employs DPC-KNN within these clustered frames to merge tokens, thereby reducing spatiotemporal redundancy. PruneVid (Huang et al., 2025c) adopts the same frame clustering methodology as Chat-UniVi. The key distinction is that it performs an initial merging of temporally static tokens before executing the spatiotemporal token consolidation. HoliTom (Shao et al., 2025) argues that relying on a single frame-level representation token for video frame clustering can lead to suboptimal detail capture, and that the preliminary merging of static temporal tokens is disconnected from the original frame clustering method. HoliTom re-conceptualizes temporal redundancy compression as an optimization problem aimed at maximizing the compressible temporal redundant features within all clustered frames, thus addressing temporal compression more holistically. DyCoke (Tao et al., 2025a) groups frames into sets of four, directly performing temporal pruning within each group.

While some methods do not explicitly cluster video frames, FrameFusion (Fu et al., 2025b), for example, acts as a token compression technique for video LLMs. It directly merges temporally redundant tokens exceeding a specific threshold in the shallow layers of the model.

## 4.3 Attention-based Video-centric Compression

Current attention-based token compression methods in video LLMs and image LLMs share significant similarities. When attention is applied within the encoder to guide token compression, videos are typically treated as a sequence of images fed into an image encoder, making these approaches similar to image-centric token compression. For a more concise discussion of such attention-based methods, please refer to Section 3.3.

In contrast, methods employing attention within the decoder process video frames sequentially, concatenating their tokens over time. For longer videos, particularly in the context of streaming video LLMs, windowed

attention is commonly used to mitigate computational overhead by focusing on local temporal visual information. However, it's notable that even these windowed attention-based methods within the decoder often share the same foundational principles as those discussed in Section 3.3.

### 4.4 Query-based Video-centric Compression

#### 4.4.1 Token Distillation

Token distillation in video LLMs commonly relies on specialized adaptor modules, such as the Q-former (Liu et al., 2023; Li et al., 2023c) or Token Turing Machines (Ryoo et al., 2023). These modules typically process video tokens with the learnable query tokens to be attended.

Token Turing Machines (TTMs) (Ryoo et al., 2023) maintain a compact external memory of summary tokens, sequentially compressing both new input tokens and memory at each timestep via a Transformer-based read/write mechanism, allowing scalable and efficient processing of long video sequences. BLIP-3-Video (Ryoo et al., 2024) introduces an explicit temporal encoder that abstracts hundreds of frame-level visual tokens into as few as 16–32 spatiotemporal tokens using learnable pooling and sequential models, enabling efficient video understanding with limited token usage. LinVT (Gao et al., 2024a) proposes a plug-and-play Linear Video Tokenizer, which linearly aggregates frame-level visual tokens into a compact set of video tokens through spatio-temporal scoring, multi-scale pooling, and text-conditioned aggregation, enabling existing image-LLMs to efficiently process videos and dynamically extract question-relevant information. Long-VMNet (Gurukar & Kadav, 2025) accelerates long-form video understanding by using a neural sampler to select discriminative visual tokens from clips and storing them in a fixed-size memory bank for each video; downstream queries are answered by processing only these memory tokens, greatly reducing computational cost while preserving key spatiotemporal information. STORM (Jiang et al., 2025a) inserts a Mamba-based (Gu & Dao, 2024a) temporal encoder between the image encoder and LLM, using spatiotemporal scanning and pooling to inject temporal context into frame tokens and then aggressively compresses tokens by temporal and spatial pooling, enabling efficient long video understanding with minimal token loss. To understand more methods and applications of token distillation in video LLMs, please also refer to Section 3.4 for a detailed explanation.

#### 4.4.2 Cross-Modal Selection

In video large language models (video LLMs), a query is commonly used to guide the selection of salient frames. In extreme cases, only a handful of frames are relevant to the posed question, allowing the tokens from the vast majority of remaining frames to be discarded. When dealing with an immense number of frames, finding query-relevant information can be akin to searching for a "needle in a haystack" for the LLM. Query-based token compression methods can pre-filter query-relevant tokens, significantly alleviating the computational burden on the LLM.

LongVU (Shen et al., 2025b) exemplifies this approach. It calculates the relevance of each video frame to the query via cross-modal interaction. This relevance score then dictates a lower compression ratio for key frames, better preserving critical information, all while ensuring the total number of tokens remains within the maximum context length of the LLM.

## 5 Audio-centric Token Compression

For audio LLMs, the demand for longer context arises from the need to process higher sampling rates and extended durations of audio.

The extraction of information from the audio modality can be categorized according to the format of audio representation: (1) **continuous sequence modeling:** this approach utilizes a pre-trained audio encoder, typically models like Whisper (Radford et al., 2023) or Conformer (Gulati et al., 2020), to produce continuous audio embeddings; (2) **discrete sequence modeling:** this method transforms the input audio signal into discrete audio tokens, usually via vector quantization, where continuous audio features are encoded into a

learnable codebook. Mainstream methods include HuBERT (Hsu et al., 2021) and EnCodec (Défossez et al., 2022; Zeghidour et al., 2021).

The second category inherently reduces the number of tokens by the design of the tokenizer structure and the codebook. Nevertheless, detailed exploration of these specific design considerations falls outside the purview of this survey.

Audio, a 1D signal representing amplitude over time, must be transformed into a suitable format for deep learning models, especially when integrating with MLLMs. MLLMs often leverage architectures designed for 2D data (like images) or general sequences. While the raw waveform is the source, spectrograms (especially Mel-spectrograms) are frequently the preferred representation for audio in MLLMs. This preference arises because spectrograms allow the application of processing techniques similar to those used for images, thereby facilitating multimodal learning.

Consequently, much like the visual modality, we categorize audio token compression methods as follows:

## 5.1 Transformation-based audio-centric Compression

Following the visual modality's categories, we can classify methods based on their downsampling operations:

### 5.1.1 Token Stacking

Similar to the pixel unshuffle operation in 2D image processing, this approach for audio LLMs token compression involves stacking multiple consecutive tokens along the hidden dimension of the token. This effectively reduces the total number of tokens. Notably, HTS-AT (Chen et al., 2022), an early example of audio token stacking for classification tasks within audio transformers, utilized 2D pixel-unshuffling on the 2D features extracted from Mel spectrograms to reduce audio tokens. More recent methods such as SLAM-ASR (Ma et al., 2024c), LLaMA-Omni (Fang et al., 2024), Llama-AVSR (Cappellazzo et al., 2025a) and others (Fathullah et al., 2024) stack the audio token. Since these token stacking operations alter the hidden dimension, an MLP is typically used to realign the dimension for compatibility with other modalities.

### 5.1.2 Pooling

Another common technique for reducing the number of audio tokens is pooling. Models like Qwen2-audio (Chu et al., 2024b) and Qwen2.5-Omni (Xu et al., 2025b) leverage pooling layers with a stride of 2 to directly decrease the length of the audio representation in a parameter-free manner. This effectively downsamples the audio features, leading to a more compact token sequence. Extending this concept, Llama-MTSK (Cappellazzo et al., 2025b) employs a matryoshka-based training approach for flexible token compression. It trains the model with multi-scale audio and video information by applying average pooling or token stacking at different rates to the initial tokens. This enables Llama-MTSK to dynamically adjust the number of tokens processed during inference, balancing compression and performance within a single model.

### 5.1.3 Temporal Convolution

For audio tokens, 1D convolutions applied across the temporal dimension can reduce the number of tokens. This method simultaneously allows for the alignment of the hidden dimension for subsequent LLM. Approaches like SpeechVerse (Das et al., 2024), Baichuan-Audio (Li et al., 2025c), OSUM (Geng et al., 2025), and LUCY (Gao et al., 2025) have employed this technique, often resulting in a downsampled audio representation with an effective sampling rate of 12.5 Hz.

These methods demonstrate how insights from image compression, particularly involving transformations, can be effectively applied to the audio domain to achieve more efficient token representation for large models.

## 5.2 Similarity-based audio-centric Compression

Similarity-based compression methods aim for each audio token to carry unique information rather than being overly redundant. Similar to the ToMe (Bolya et al., 2022) method used in vision transformers (ViT),

A-ToMe (Li et al., 2023f) inserts a token merge module between the multihead self-attention (MHSA) and feed-forward network (FFN). This module merges adjacent audio tokens that have high cosine similarity.

### 5.3 Attention-based audio-centric Compression

For audio tasks, attention-based methods are also effectively utilized to compress tokens.

#### 5.3.1 Attention in Encoder

Top-K (Lee & Lee, 2025) is a token selection method operating within the audio spectrogram transformer block. It retains only the top K audio tokens ranked by the magnitude of their attention scores. This prunes less attentive tokens, focusing on those with higher relevance as determined by the self-attention mechanism.

#### 5.3.2 Attention in Decoder

SpeechPrune (Lin et al., 2025b), works in the LLM backbone. It prunes audio tokens based on attention scores provided by the first transformer layer. By utilizing the initial layer's attention, SpeechPrune efficiently identifies and discards less crucial tokens early in the processing pipeline, aiming to reduce computational load and improve efficiency for subsequent layers without significant loss of information.

### 5.4 Query-based audio-centric Compression

Audio feature representations can also be compressed using other modalities or learned query mechanisms. Analogous to image LLMs, these methods can be broadly categorized into token distillation and cross-modal selection, based on whether learned queries are explicitly employed.

#### 5.4.1 Token Distillation

This category leverages learnable query tokens to distill comprehensive audio information into a compact, fixed-length representation.

Video-LLaMA (Zhang et al., 2023a) and SALMONN series (Tang et al., 2024; Sun et al., 2024b) employ an audio Q-former to transform variable-length audio inputs into a fixed-length sequence of learnable queries, thereby condensing audio information for the LLM. MMCE-Qformer (Xue et al., 2024) compresses acoustic information by utilizing learnable queries to extract global acoustic context from contextual audio embeddings. Concurrently, a cross-attention mechanism, guided by input text embeddings, captures local acoustic context relevant to each text token. This dual approach distills both broad and specific audio features into compact, text-relevant representations. MMS-LLaVA (Yeo et al., 2025) reduces multimodal token length for efficient speech LLMs. It first halves the sequence length with an Early AV-Fusion Module, which combines visual and audio features. Subsequently, an AV Q-Former further compresses these fused features into a fixed number of queries, effectively capturing full speech context to bridge the token gap with text.

#### 5.4.2 Cross-Modal Selection

Similar to the visual modality, audio token compression can also be guided by information from other modalities. Speechprune (Lin et al., 2025b), for example, leverages audio-text correlation to identify semantically important audio segments. This is achieved by calculating a cross-modal similarity matrix based on cosine similarity, which then guides the compression of audio tokens. This approach ensures that the most relevant audio information is retained.

### 5.5 Discussion about Specific Redundancy of Audio

Distinct from visual modalities, audio signals exhibit high sampling rates and significant spectro-temporal correlations. Even brief speech segments yield hundreds of tokens, a substantial portion of which encapsulate overlapping or redundant information. This section delineates redundancy patterns inherent to au-

dio—specifically spectral redundancy, temporal redundancy, and silence or repetitive noise—to establish a foundation for efficient token compression in audio LLMs.

### 5.5.1 Spectral and Temporal Redundancy

Like video, audio exhibits intrinsic temporal structure. Consequently, compressing tokens along the temporal dimension is a well-founded strategy (Someki et al., 2025). Concurrently, given that the high sampling rate of audio generates dense token sequences that burden computational efficiency, it is imperative to mitigate spectral redundancy while preserving semantic integrity. Recently, Bhati et al. (2025) pioneered token pruning for audio LLMs by utilizing spectral features for segmentation before addressing temporal redundancy. Their method achieves a substantial reduction in token density with minimal fine-tuning requirements.

### 5.5.2 Silence and Audio Noise

Many ASR pipelines explicitly remove long pauses and noise, effectively performing coarse-grained pruning at the waveform level. Nevertheless, end-to-end systems still receive audio-token sequences with muted or noisy segments. Although some tokens are redundant, others carry contextual cues beneficial to downstream tasks; consequently, developing principled audio-token pruning remains a promising yet challenging avenue for future work.

## 6 Discussions

### 6.1 Synergies and Distinctions with Other Compression Methods

Beyond token compression, the research community has seen the emergence of several other compression methods, including model quantization (Lin et al., 2024b; Xiao et al., 2023a; Frantar et al., 2023; Shang et al., 2023; Sui et al., 2024a; Gholami et al., 2022), network pruning (Han et al., 2016; Ma et al., 2023; Sui et al., 2021; Cheng et al., 2024a), knowledge distillation (Hinton et al., 2015; Gou et al., 2021), and low-rank factorization (Yu et al., 2017; Yin et al., 2021; Xiao et al., 2023b; Sui et al., 2024b; Yang et al., 2024). These methods typically focus on directly compressing model weights to achieve efficiency.

For Transformer-based models, the computational cost (FLOPs) is mainly dominated by matrix multiplications, particularly in the self-attention and feed-forward layers. A simplified formulation is given as:

$$\text{FLOPs} \propto O(N \cdot D^2 + N^2 \cdot D), \tag{11}$$

where $N$ is the number of tokens, $D$ is the model dimension.

### 6.1.1 Weight-Focused Compression Methods

These methods mainly target the model dimension ($D$) by reducing the effective size or complexity of the model weights. **Model Quantization** reduces weight precision, directly impacting the memory associated with $D$. A key limitation is that highly aggressive quantization (e.g., 4-bit) often compromises accuracy, meaning there's no "free lunch" when it comes to achieving lossless performance. Furthermore, effectively accelerating these lower bit-rates often necessitates specialized hardware. **Network Pruning** removes redundant connections, effectively reducing the active parameters contributing to $D$. For LLMs, aggressive structured pruning (e.g., beyond 20% for downstream tasks) often leads to significant performance degradation or near-collapse due to the difficulty in preserving architectural integrity. **Knowledge Distillation** trains a smaller student model (with a smaller $D$) to mimic a larger teacher (Hinton et al., 2015). Its main limitation is the "knowledge gap", as the student may struggle to fully capture the teacher's comprehensive knowledge, leading to performance disparities, especially on complex or out-of-distribution data. **Low-Rank Factorization** decomposes weight matrices into lower-rank approximations, thus reducing parameters related to $D$. The challenge lies in finding an optimal low-rank approximation for diverse tasks without performance loss, as this is often task-dependent and complex to apply consistently across deep networks.

### 6.1.2 Token Compression

In contrast, token compression directly targets the sequence length ($N$) by reducing the number of tokens processed for long contexts. By reducing $N$, token compression significantly impacts FLOPs:

$$\text{FLOPs} \propto O(M \cdot D^2 + M^2 \cdot D), \tag{12}$$

where $M \ll N$ represents the reduced sequence length after token compression.

This approach offers benefits like greater efficiency for long context processing, overcoming context window limitations, and closer alignment with API cost reduction, as many LLM APIs charge by token count.

### 6.1.3 Complementary Nature and Synergistic Gains

The methods for compressing model weights and token compression are structurally orthogonal and can be effectively combined for superior results. For example: NVILA (Liu et al., 2025e) pushes inference latency reduction and throughput maximization to the extreme by simultaneously applying quantization and token compression. CoreMatching (Wang et al., 2025b) achieves synergistic acceleration by concurrently compressing both neurons (a form of pruning/weight reduction) and tokens.

This orthogonality means that combining these approaches holds the potential for compounded efficiency gains that are greater than applying either method in isolation.

## 6.2 Token Compression: Efficiency and Beyond

Token compression is often perceived solely as a training-free method to boost efficiency. However, its significance extends far beyond this, having been intrinsically incorporated into the design of MLLM, particularly within the modality transition modules (*e.g.*, adapter). This integration not only facilitates superior modality alignment but also enhances the quality of information, leading to more efficient and stable training.

### 6.2.1 Enhanced Modality Alignment

Effectively aligning and comprehending information from disparate modalities remains a significant challenge. Traditional encoders segment and tokenize all multimodal information to align with linguistic representations. However, low-quality and low-density multimodal representations expand the alignment space, complicating the task of modality matching. Token compression addresses this by enabling a more precise correspondence between language representations and multimodal information.

A prime example is the Q-Former (Liu et al., 2023; Li et al., 2023c), which employs a trainable vector to distill visual tokens, achieving direct alignment of the modality simultaneously. Similarly, $M^3$ (Cai et al., 2024a) adopts a coarse-to-fine semantic granularity training approach, empowering MLLMs to align with and interpret visual representations at various levels.

### 6.2.2 Improved Information Representation

The sheer volume of multimodal information often leads to inefficient training and inference, with an overabundance of multimodal tokens that potentially degrade the capabilities of the text modality (Bellver-Soler et al., 2025). This issue is compounded by inherent redundancies within multimodal data itself: **(1) Feature Redundancy** arises from similar backgrounds in visual data or silent segments in audio. **(2) Task-Irrelevant Redundancy** is evident in tasks like visual question answering (VQA), where a significant portion of multimodal representations may be irrelevant to deriving the correct answer. **(3) Attention Computation Redundancy** emerges from two aspects: first, due to the nature of attention mechanisms, tokens positioned later in a sequence often receive disproportionately higher attention (Wen et al., 2025), suggesting potential computational redundancy for tokens not at the sequence's end; and second, because multimodal information receives inherently less attention than textual data (Chen et al., 2024a; Song et al., 2025a), an abundance of multimodal tokens can still introduce substantial computational redundancy.

Addressing these issues, the method classifications discussed earlier directly correspond to these types of data redundancy. Specifically, the transformation-based methods along with similarity-based approaches,

are effective in mitigating the feature redundancy. Furthermore, attention-based methods play a crucial role in minimizing attention computation redundancy. Lastly, query-based methods are designed to reduce task-irrelevant redundancy.

### 6.2.3 Enable One-Shot Long-Context Understanding

Limited by the inherent length of the context, MLLMs are unable to comprehend real-world scenarios involving extremely long contexts, such as understanding entire code repositories or extended video and audio sequences (Qu et al., 2025). However, token compression significantly condenses and abstracts original information representations, making it possible for MLLMs to understand these long contexts in a single pass.

Traditional methods for handling long contexts in MLLMs, like FlashAttention (Dao et al., 2022; Dao, 2024) or RingAttention (Liu et al., 2024a), involve architectural changes to the model's attention mechanism to directly accommodate longer sequences. While effective, these require fundamental model modifications. Token compression offers a different, often simpler, route. Instead of redesigning the model to fit more tokens, it focuses on making each token more powerful. By creating information-dense tokens, we pack more meaning into fewer pieces of data. This lets existing MLLM architectures process significantly longer conceptual contexts without major overhauls. It's a more efficient and accessible way to achieve that crucial one-shot understanding of vast, complex real-world information (Song et al., 2025b).

## 6.3 Combining Different Token Compression Methods

In Section 6.2.2, we explored three distinct types of redundancy and the corresponding methods to reduce them. This raises a natural question: can we combine multiple token compression methods to achieve a synergistic effect?

We observe that while certain approaches operate orthogonally, others may exhibit conflicts.

For instance, we can first eliminate structural redundancy by addressing feature redundancy, and subsequently filter out task-irrelevant redundancy by selecting tokens most pertinent to the user query. Since these strategies address distinct dimensions of the data, this combination is fundamentally orthogonal.

Similarly, strategic combinations can yield superior performance through careful design. VisionZip (Yang et al., 2025c), for example, prioritizes tokens with high attention scores in the ViT to preserve critical information, while consolidating the remaining tokens via similarity-based merging. This approach safeguards key features from being diluted by similarity-based aggregation. Although these methods are not strictly orthogonal, a tailored design enables them to complement each other effectively.

Conversely, certain combinations may conflict, such as pairing external query-based pruners with attention-based selection in the decoder. Since the decoder's cross-attention naturally acts as a text-guided filter for multimodal tokens, applying an external query-based compressor beforehand often yields diminishing returns. This occurs because the specific information required to answer a query dictates a lower bound on the token count, limiting the potential for further compression.

## 6.4 Cross-modal token compression

For the joint compression of token across modalities, the prevalent paradigm utilizes the textual modality to compress visual or audio representations. This approach underpins the vast majority of query-based attention methods (See Section 3.4, 4.4, and 5.4).

Conversely, some approaches leverage the visual modality to guide text token compression; for instance, SparseVLM (Zhang et al., 2024c) employs mutual supervision between text and visual modalities to compress tokens in both. Additionally, OmniZip (Tao et al., 2025b) introduces a "listen-to-prune" mechanism, utilizing audio cues to jointly guide the compression of audio and video tokens. Furthermore, given the inherent and distinct redundancies within each modality, orthogonal compression strategies can be stacked to further

maximize token reduction. To the best of our knowledge, literature exploring strategies beyond text-guided compression remains scarce; consequently, cross-modal joint optimization represents a promising direction for future research.

## 6.5 Current Challenges

### 6.5.1 Performance Degradation

While token compression can effectively condense multimodal features, it also introduces a risk of performance degradation. Current research on visual MLLMs, for example, shows that for models like LLaVA-OV-7B (Li et al., 2025a), near-lossless performance can be achieved by retaining as few as 10% of the original tokens. However, performance declines sharply when the compression rate is pushed further. This challenge is more pronounced for larger and more recent models such as Qwen2.5-VL (Bai et al., 2025), LLaVA-Video-7B (Zhang et al., 2024d), and LLaVA-OV-72B (Li et al., 2025a), where achieving lossless compression seems to be more difficult.

This increased difficulty may stem from the models' enhanced representational capabilities. It has been suggested that less capable models are inherently less sensitive to information loss from aggressive compression, as their weaker understanding already struggles to process the complex, uncompressed data fully. In contrast, more sophisticated models, which possess a more nuanced and holistic comprehension of multimodal tokens, are more susceptible to the subtle degradation caused by compression. For these models, achieving high performance requires a far more delicate and precise approach to preserve the token.

### 6.5.2 Task-Specific Challenges

Token compression, while beneficial for efficiency, can be destructive to performance on tasks that demand high representational fidelity. For **optical character recognition (OCR)**, which requires a high information density within local regions, compression often leads to the loss of critical details and a subsequent drop in performance. This is particularly evident on benchmarks like RefCOCO (Yu et al., 2016), where the model's ability to ground objects based on fine-grained textual cues is compromised.

A similar challenge arises in preserving **temporal perception**. Video and audio are fundamentally structured by fixed sampling rates (Liu et al., 2025d). By merging adjacent frames or sequential tokens, compression methods disrupt this inherent temporal consistency, hindering the model's ability to reason about motion, pace, and other crucial temporal dynamics essential for a complete understanding of the content.

### 6.5.3 Deployment Hurdles

Despite their potential, many token compression methods face barriers to real-world deployment, stemming from a fundamental incompatibility with current large-scale model architectures and applications.

A major challenge lies in their integration with modern acceleration libraries (Dao et al., 2022; Dao, 2024). Methods that rely on explicit attention scores to prune tokens cannot be seamlessly integrated into *current optimized frameworks*, as these libraries fuse matrix multiplication and softmax operations to maximize throughput and minimize memory usage, thus making those scores inaccessible. This creates a critical gap, as these compression methods cannot leverage the performance gains of state-of-the-art deployment pipelines.

Furthermore, task-aware token compression methods are not suit for *multi-turn conversational tasks*. Methods that perform token compression internally within the model's backbone or rely on cross-modal fusion are not natively compatible with this type of application. They lack an efficient mechanism to carry over and update a compressed representation across turns, instead requiring a costly re-computation of the entire conversation history for each new query.

### 6.5.4 Evaluation Challenges

**Rethinking Evaluation Metrics.** Current evaluation methods for token compression techniques face limitations, hindering accurate and comprehensive comparisons.

Table 4: Common Benchmarks for Performance Evaluation of Image-Language and Video-Language Tasks.

| Benchmark | Task | Metric | System Prompt |
|---|---|---|---|
| *Image Task* | | | |
| GQA (Hudson & Manning, 2019) | CE-VQA | Exact Match | Answer the question using a single word or phrase. |
| MMB (Liu et al., 2024d) | MC-VQA | Accuracy | Answer with the option's letter from the given choices directly. |
| MME (Yin et al., 2024) | CE-VQA | Perception Score | Answer the question using a single word or phrase. |
| POPE (Li et al., 2023e) | CE-VQA | F1 Score | Answer the question using a single word or phrase. |
| ScienceQA-Image (Lu et al., 2022) | Visual reasoning | Exact Match | Answer with the option's letter from the given choices directly. |
| SeedBench-Image (Li et al., 2023a) | MC-VQA | Accuracy | Answer with the option's letter from the given choices directly. |
| VizWiz (Gurari et al., 2018) | CE-VQA | Exact Match | When the provided information is insufficient, respond with "Unanswerable". Answer the question using a single word or phrase. |
| VQA$^2$ (Goyal et al., 2017) | CE-VQA | Exact Match | Answer the question using a single word or phrase. |
| MM-Vet (Yu et al., 2023) | Visual reasoning | GPT-score | First please perform reasoning, and think step by step to provide the best answer to the following question: |
| LLaVA$^W$ (Liu et al., 2023) | Visual reasoning | GPT-score | A chat between a curious human and an artificial intelligence assistant. The assistant gives helpful, detailed, and polite answers to the human's questions. |
| *Video Task* | | | |
| ActivityNet (Yu et al., 2019) | CE-VQA | Accuracy / GPT-score | Answer the question using a single word or phrase. |
| VideoChatGPT (Maaz et al., 2023) | OE-VQA | GPT-score | Evaluate the temporal accuracy of the prediction compared to the answer.* |
| NextQA (Xiao et al., 2021) | CE-VQA | WUPS | Answer a question using a short phrase or sentence. |
| EgoSchema (Mangalam et al., 2023) | MC-VQA | Accuracy | Answer with the option's letter from the given choices directly. |
| MVBench (Li et al., 2024a) | MC-VQA | Accuracy | Carefully watch the video and pay attention to the cause and sequence of events, the detail and movement of objects, and the action and pose of persons. Based on your observations, select the best option that accurately addresses the question. |
| LongVideo Bench (Wu et al., 2024) | MC-VQA | Accuracy | Answer with the option's letter from the given choices directly. |
| VideoMME (Fu et al., 2025a) | MC-VQA | Accuracy | Select the best answer to the following multiple-choice question based on the video and the subtitles. Respond with only the letter (A, B, C, or D) of the correct option. |
| PerceptionTest (Patraucean et al., 2024) | MC-VQA | Accuracy | Answer with the option's letter from the given choices directly. |
| VideoDC (LMMs-Lab, 2024) | Video Caption | GPT-score | Please provide a detailed description of the video, focusing on the main subjects, their actions, and the background scenes |
| AuroraCap (Chai et al., 2025) | Video Caption | VDCscore | Describe the video in detail. |
| Chardes-STA (Gao et al., 2017) | Temporal Grounding | IoU | Please find the visual event described by a sentence in the video, determining its starting and ending times. |

For methods requiring training, various factors like training data and methodologies make it challenging to isolate and directly compare the effectiveness of different methods.

For training-free token compression methods, current evaluations often rely on metrics such as the number of compressed tokens and FLOPs. However, these metrics offer an incomplete picture. While the number of compressed tokens provides a preliminary classification, the compression location significantly impacts the downstream computational load; earlier pruning generally leads to greater reductions. Similarly, FLOPs, while useful for theoretical computational estimates, frequently do not accurately reflect actual inference speed. Therefore, for training-free methods, more practical metrics like Time To First Token (TTFT) and decoding latency per token are crucial for a more accurate assessment of real-world inference acceleration.

**Evaluation Benchmarks Gap.** Current evaluation datasets for MLLM token compression often rely on general multimodal benchmarks (Table 4), provide insufficient granularity. For example, in challenging long video understanding tasks, performance hinges more on sparse frame sampling, capturing key frames, than on the specific token compression method. This can obscure the true impact of token compression, making

its efficacy appear negligible. Furthermore, relying solely on VQA datasets that demand only low-fidelity information is insufficient, as they lack the fine-grained sensitivity required to evaluate token compression.

This reveals a critical gap: current datasets often fail to isolate and precisely measure the effect of token compression. Therefore, adopting specific designed evaluation methodologies like EffiVLM-Bench (Wang et al., 2025c), VTC-Bench (Liao et al., 2025) and challenging benchmarks such as OCR (Yu et al., 2016) and temporal grounding (Gao et al., 2017) benchmarks, is crucial for accurately assessing the true efficacy and nuanced benefits of token compression methods.

### 6.6 Pruning Location and Trade-offs

Given the cascaded architecture of current MLLMs, the placement of the pruning operation directly influences the trade-off between computational efficiency and performance.

Pruning tokens at an early stage, such as within the encoder or projector, can dramatically shorten the sequence length. This significantly reduces the computational burden on the downstream LLM, leading to faster inference. However, this early compression carries a higher risk of discarding critical information, which can negatively impact model performance.

Conversely, token compression at a later stage, within the LLM's internal modules, is more computationally demanding. However, it reduces the risk of erroneous judgment because the tokens have already undergone initial processing and feature extraction, thereby retaining more refined information. The optimal location for token compression within these architectures remains an open question, warranting further investigation.

### 6.7 Future Directions

#### 6.7.1 Joint Token Compression for Multimodal Settings

While distinct modalities exhibit unique redundancy patterns requiring specialized handling, the field is rapidly evolving towards Omnimodal Large Language Models (omni LLMs) capable of real-time, joint inference (Xu et al., 2025b; Tong et al., 2025; Xu et al., 2025c; Xie & Wu, 2024; Tang et al., 2025; Yang et al., 2025b; Ge et al., 2025; Fu et al., 2024; Shu et al., 2025a; Sun et al., 2024a; Li et al., 2024c). However, single-modal deployments remain constrained by their unimodal inputs. As established in Sections 3, 4, and 5, fundamental algorithmic principles (including transformation-based, similarity-based, attention-based, and query-based approaches) demonstrate transmodal applicability, indicating the viability of developing a unified multimodal token compression framework. A promising future direction lies in exploiting **cross-modal synergy** to reduce the aggregate token count. Pioneering efforts like OmniZip (Tao et al., 2025b) have begun to explore this by utilizing audio cues to guide visual pruning, underscoring the predictive utility of one modality over another. Future research should further investigate deep joint compression mechanisms, where the redundancy of audio, video, and textual tokens is evaluated holistically, enabling efficient, long-context interaction for next-generation omni LLMs.

#### 6.7.2 Improved Architecture

Current token compression methods are often employed as a remedial measure to process long contexts efficiently. However, a more valuable approach might involve designing model architectures that intrinsically account for data redundancy during their initial conception. By doing so, the number of tokens could be reduced during the abstraction of data features. This is particularly relevant for current architectures, especially those of video LLMs, where generated tokens still exhibit significant redundancy. Therefore, exploring architectural designs that inherently foster more condensed information abstraction from the outset represents a promising research direction.

Furthermore, recent architectures utilizing linear attention (Gu & Dao, 2024b; Peng et al., 2023; Sun et al., 2023; Qiu et al., 2025) have emerged as a parallel solution, mitigating the computational explosion associated with increasing token counts through linear complexity. However, determining how to effectively identify and eliminate input redundancy within these novel frameworks to achieve more compact data representations remains a promising avenue for future exploration.

# 7 Applications

The potential of multimodal token compression extends beyond technical enhancements, emerging as a universal efficiency engine for data-intensive AI systems. Multimodal models frequently process extreme-length token sequences exhibiting high task-agnostic redundancy according to empirical analyses. Capitalizing on recent breakthroughs, we delineate four high-impact application domains:

## 7.1 GUI Agents and Human-Computer Interaction

Graphical user interface (GUI) agents perceive and interact with visual interfaces, interpret natural language instructions, analyze GUI states, and execute corresponding actions. These agents have to parse screen streams in real-time, producing extensive token sequences that often exceed computational limits (Zhang et al., 2024b; Wang et al., 2024a). Multimodal token compression enhances the efficiency of GUI agents. This approach mitigates context overflow in extended operation sequences by dynamically compressing redundant visual elements (e.g., extra white space or simple backgrounds). For some small but important control elements, it should also eliminate other irrelevant visual elements and highlight their importance. For instance, ShowUI (Lin et al., 2025a) is the first model to apply token selection strategy to GUI agents. ShowUI segments GUI screenshots into connected components by clustering pixels with similar RGB values, significantly reducing the total number of discrete elements. During both training and inference phases, the system employs an adaptive token selection strategy that probabilistically prunes redundant tokens within these components, thereby optimizing computational efficiency while preserving functional semantics However, excessive compression risks inducing operational ambiguity, necessitating careful calibration.

## 7.2 Healthcare and Medical Imaging

The effective synthesis of multimodal medical data is pivotal to advancing contemporary medical diagnosis and research. MLLMs can integrate radiographic findings, medical histories, and ancillary diagnostic tests to generate differential diagnoses, which clinicians can correlate with patient records and physician notes to enhance diagnostic accuracy (Liang et al., 2024). Furthermore, MLLMs can automatically draft preliminary radiology reports, potentially reducing the workload of radiologists (Beddiar & Oussalah, 2023; Bazi et al., 2023; He et al., 2020). A major challenge for MLLMs in medical imaging is the processing of high-resolution images, such as Whole-slide Images (WSIs) in pathology, which can contain billions of pixels. To overcome this, TCP-LLaVA (Lyu et al., 2025) uses a set of trainable compression tokens to aggregate and condense crucial information from thousands of visual and textual inputs. Instead of feeding every single image patch token into the language model, only these compressed tokens are forwarded for answer generation. Token compression holds vast potential for widespread application in the field of healthcare and medical imaging, enabling the efficient analysis of complex, high-resolution images.

## 7.3 Robotics and Autonomous Systems

Leveraging the significant capabilities of video LLMs in long-form video comprehension enables their deployment in robotics (Wei et al., 2025) and autonomous driving systems (Ma et al., 2024b; Zhou et al., 2024; Zhu et al., 2025b). However, the inherent computational complexity of long-duration video processing creates fundamental latency-efficiency tradeoffs that challenge real-time implementation. Token compression addresses this by prioritizing salient spatio-temporal dynamics (e.g., agent movements, action trajectories) and fine-grained per-frame details, enabling computationally efficient video understanding for these domains. VTS (Ma et al., 2024b) proposes a token pruning strategy for autonomous driving scenarios. VTS employs a proposal model based on a lightweight convolutional neural network that is able to adaptively identify keyframes and pry less informative tokens (e.g., invariant backgrounds and stationary objects). StreamVLN (Wei et al., 2025) further enhances inference efficiency for real-time navigation by employing a voxel-based spatial pruning strategy at test time to reduce memory tokens. This approach makes real-time navigation feasible.

### 7.4 Efficient Reasoning

Token compression improves efficiency by removing redundant input tokens. However, in many cases, the main source of computational cost shifts from input to output, most notably in reasoning models (Team et al., 2025; Guo et al., 2025a; Jaech et al., 2024), where lengthy generation chains are common. The "slow-thinking" paradigm improves reasoning ability but results in lengthy reasoning chains (Feng et al., 2025a; Sui et al., 2025; Chen et al., 2025; Feng et al., 2025c;b). Some efficient reasoning methods compress these chains using similar techniques (e.g., attention mechanisms, semantic importance) (Ma et al., 2025a; Xia et al., 2025; Liu et al., 2024b; Fang et al., 2025), typically requiring fine-tuning via Supervised Fine-Tuning (SFT) or Reinforcement Learning (RL). Beyond token compression, other approaches improve reasoning efficiency by compressing model (Magister et al., 2022; Li et al., 2023b; Feng et al., 2024; Zhang et al., 2025e) or accelerating decoding (Sun et al., 2024c; Ma et al., 2024a; Luo et al., 2025a; Xu et al., 2025a; Ding et al., 2025).

## 8 Conclusion

This paper presents the first structured survey of token compression techniques for Multimodal Large Language Models (MLLMs), establishing a taxonomy based on modality-specific redundancy and underlying compression mechanisms. While current methods demonstrate promising efficiency gains, several critical challenges remain on the path toward scalable and robust MLLMs. Future research must move beyond simple redundancy reduction to address the preservation of cross-modal alignment under high compression ratios and the maintenance of causal reasoning capabilities in temporal sequences. Furthermore, the field necessitates the development of specialized benchmarks designed to rigorously evaluate multi-frame comprehension and long-term context retention. We hope this survey serves as a roadmap, guiding the community to tackle these open problems and push the boundaries of processing increasingly complex multimodal data.

## 9 Acknowledgment

This paper is supported by Young Scientists Fund of the National Natural Science Foundation of China (NSFC) (No. 62506305), Zhejiang Leading Innovative and Entrepreneur Team Introduction Program (No. 2024R01007), Key Research and Development Program of Zhejiang Province (No. 2025C01026), Scientific Research Project of Westlake University (No. WU2025WF003), Chinese Association for Artificial Intelligence (CAAI) & Ant Group Research Fund - AGI Track (No. 2025CAAI-ANT-13). It is also supported by the research funds of National Talent Program and Hangzhou Municipal Talent Program.

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
