# OpenReview forum: "A Survey of Token Compression for Efficient Multimodal Large Language Models"
_TMLR — Accepted by TMLR_

### Review · Reviewer_LJku · 2025-10-29

**Summary Of Contributions:**

This paper presents a survey of token compression methods for multimodal long-context models, covering images, video, and audio. Its main contributions are:

- It identifies and motivates the problem of token explosion in multimodal large language models (MLLMs), explaining how input modalities like high-resolution images, long-duration video, and audio generate massive numbers of tokens which make self-attention expensive.

- It proposes a modality-aware taxonomy: splitting approaches by modality (image-centric / video-centric / audio-centric), which helps tailor compression methods to redundancy patterns specific to each modality.

- Beyond modality, it also classifies methods by mechanism: “transformation-based”, “similarity-based”, “attention-based”, and “query-based” approaches.

- It discusses the strengths, limitations, and practical trade-offs of many recent token-compression techniques under those dimensions, and identifies open challenges for future research.

- It maintains a public repository (“Awesome Multimodal Token Compression”) that tracks relevant papers and code, making the survey a living resource.

**Audience:**

Yes

**Audience Explanation:**

The survey is:

- Timely — because token compression for multimodal long contexts (especially video and audio) is now a key scalability bottleneck.

- Comprehensive — it covers multiple modalities and mechanisms, giving a wide view.

- Well-organized — the taxonomy is clear and useful for researchers entering the area.

- Practical — the GitHub repository adds community value beyond the text.

These make the paper a valuable reference even if it introduces no new algorithms.

**Claims And Evidence:**

Yes

**Claims Explanation:**

After reading the paper, its main novelty is conceptual — it organizes scattered research into a systematic taxonomy for multimodal token compression (image/video/audio) and links those with mechanism types (transformation, similarity, attention, query-based).
It also provides a public repository that makes the survey actionable and continuously updated.

Hence, my summary focused on those aspects:

- Identification of the token explosion problem in multimodal contexts.

- Introduction of a dual taxonomy (by modality and mechanism).

- Consolidation of methods and insights into one structured framework.

- Maintenance of a living repository for reproducibility.

These are the genuine contributions that differentiate this paper from earlier, more general token compression or long-context surveys.

**Requested Changes:**

- Add quantitative comparison or standardized metrics.

   - Currently, the paper provides a qualitative taxonomy but lacks quantitative evaluation (e.g., compression ratio vs. performance trade-offs across modalities).

  - Even a small unified comparison table summarizing the relative efficiency and accuracy impact of representative methods would significantly strengthen its analytical rigor.

- Clarify taxonomy boundaries and inclusion criteria.

  - The paper should explicitly define what qualifies as “token compression” versus related notions (e.g., frame sampling, key-frame extraction, patch merging, or attention sparsity).

- Expand coverage of cross-modal compression strategies.

  - The current survey mostly treats modalities separately. It should also highlight methods that jointly optimize across modalities (e.g., vision–language, audio–video compression).

- Strengthen the discussion on future challenges and open problems.

The conclusion section should move beyond summarization and provide a critical perspective — such as challenges in maintaining alignment under compression, causal temporal reasoning, or benchmarks for multi-frame comprehension.

---

> ### Author Response · Authors · 2025-12-06
> **Response**
>
> Thank you so much for the detailed and constructive comments. We address your concerns as follows.
>
> ---
>
> > **W1**: Add quantitative comparison or standardized metrics. Currently, the paper provides a qualitative taxonomy but lacks quantitative evaluation (e.g., compression ratio vs. performance trade-offs across modalities).Even a small unified comparison table summarizing the relative efficiency and accuracy impact of representative methods would significantly strengthen its analytical rigor.
>
> **Response to W1**: We thank the reviewer for the constructive suggestion. While Tables 2 and 3 in our original manuscript provided quantitative evaluations for image and video modalities (using token counts or retention ratios for computational cost), we agree that a comparison is essential to highlight the trade-offs. To fully address your concern and strengthen the analytical rigor, we have added a new Figure 4. This figure explicitly visualizes the efficiency-performance trade-offs across representative methods, offering the clearer unified perspective requested.
>
> ---
>
> > **W2**: Clarify taxonomy boundaries and inclusion criteria.
> 	The paper should explicitly define what qualifies as “token compression” versus related notions (e.g., frame sampling, key-frame extraction, patch merging, or attention sparsity).
>
> **Response to W2**: We are grateful for the opportunity to clarify our scope. We have revised Section 2.4 to specify that *token compression* pertains exclusively to methods that operate on feature embeddings within network layers to reduce the token count from $N$ to $M$ ($M < N$). By defining the boundary at the embedding level, we explicitly exclude input-level strategies (such as frame sampling) and attention-masking techniques (such as sparse attention) from our taxonomy.
>
> ---
>
> > **W3**: Expand coverage of cross-modal compression strategies. The current survey mostly treats modalities separately. It should also highlight methods that jointly optimize across modalities (e.g., vision–language, audio–video compression).
>
> **Response to W3**: We thank the reviewer for this valuable suggestion. We agree that covering cross-modal synergies is essential for a complete survey. In the revision, we have introduced a new Section 6.4 (Cross-Modal Compression Strategies). This section specifically reviews methods that exploit inter-modal correlations for joint optimization (e.g., vision–language and audio–video compression), thereby providing a more integrated and comprehensive perspective.
>
> ---
>
> > **W4**: Strengthen the discussion on future challenges and open problems.
>
> **Response to W4**: We are grateful for this insightful suggestion. We agree that the Conclusion should offer a critical outlook rather than merely summarizing the contributions. Accordingly, we have substantially revised this section to highlight key open problems, specifically addressing the challenges. We believe these additions provide a more valuable and inspiring roadmap for future research.

---

### Review · Reviewer_WCxP · 2025-11-27

**Summary Of Contributions:**

This paper presents the a systematic survey on **multimodal long-context token compression**, addressing the computational bottlenecks in MLLMs caused by the long context generated from high-resolution images, long videos, and audio. The authors propose a structured taxonomy that categorizes existing methods first by **modality** (image, video, and audio) to address specific redundancies, and further by their **underlying mechanisms**, such as transformation-based, similarity-based, attention-based, and query-based approaches. Additionally, the work consolidates current progress, analyzes key challenges like performance degradation and deployment incompatibility, and outlines future research directions for efficient multimodal understanding.

Strengths:
1. With the rapid development of MLLMs, addressing the efficiency bottleneck of long context is an important research area. This survey introduces the methods of multimodal token compression in detail.
2. The taxonomy by both modality and algorithmic mechanism makes the survey highly structured for researchers looking for specific solutions.
3. The discussion on deployment hurdles and evaluation challenges is practical and good.

Weakness:
1. These is no discussion on long-context compression methods in LLMs. The text modality is also one important modality. There are many important context compression methods, which inspire other methods in the multi-modal area.
2. Miss the discussion on methods using sparse attention. The attention-based compression methods are similar to some works in sparse attention. A relevant discussion can make this survey better.
3.  The discussion on linear attention is also important for multi-modal context compression. Especially, the recent Qwen3-Next has proven its potential and ability.

**Audience:**

Yes

**Audience Explanation:**

There is likely high interest in this work for several reasons. Firstly, MLLMs are currently a popular topic. As these models scale to handle long videos and high-resolution images, the long context problem affects almost all practitioners and researchers working on efficient inference and training. Many researchers in the TMLR community operate with resource constraints. A survey detailing methods to reduce memory and compute costs (e.g., reducing 90-minute videos from millions of tokens to manageable sizes ) is highly practical and valuable.

**Claims And Evidence:**

Yes

**Claims Explanation:**

The submission is a survey and taxonomy proposal rather than an empirical study of a new method. In this context, this survey has be well supported by lots of existing methods. The authors clearly justify their taxonomy by analyzing the inherent redundancy types in different modalities and different compression methods, supported by references to foundational works in ViTs and MLLMs. The paper provides extensive tables that aggregate performance metrics (like VQA accuracy) and token retention ratios from the cited papers. This supports their analysis of the trade-offs between compression rates and performance preservation.

**Requested Changes:**

1. Include discussions on long-context compression methods in LLMs, such as H2O [1], StreamingLLM [2], SnapKV [3], PyramidKV [4], and so on.

[1] H2O: Heavy-Hitter Oracle for Efficient Generative Inference of Large Language Models (Zhang et al., NeurIPS 2023).

[2] Efficient Streaming Language Models with Attention Sinks (Xiao et al., ICLR 2024).

[3] SnapKV: LLM Knows What You are Looking for Before Generation (Li et al., 2024).

[4] PyramidKV: Dynamic KV Cache Compression for Long-Context LLMs (2024).

2. Include discussions on methods that use sparse attention, such as Longformer[5], shadowAttn[6], SpargeAttention[7], Deepseek's NSA[8], MoBA[9], and so on.

[5] Longformer: The Long-Document Transformer.

[6] shadowAttn: Dynamic Sparse Attention on Mobile SoCs

[7] SpargeAttention: Accurate and Training-free Sparse Attention Accelerating Any Model Inference

[8] Native Sparse Attention: Hardware-Aligned and Natively Trainable Sparse Attention

[9] MoBA: Mixture of Block Attention for Long-Context LLMs


3. Include discussions on works that utilize linear attention, such as  Mamba[10] , RWKV[11], RetNet[12], Qwen3-Next(Gated Attention[13]), and so on.

[10] Mamba: Linear-Time Sequence Modeling with Selective State Spaces

[11] RWKV: Reinventing RNNs for the Transformer Era

[12] Retentive Network: A Successor to Transformer for Large Language Models

[13] Gated Attention for Large Language Models: Non-linearity, Sparsity, and Attention-Sink-Free

4. Some important works on multi-modal context token compression are missing. Please add the discussion on recent EMLoC[14] and previous Token Merger[15].

[14] Efficient Multi-modal Long Context Learning for Training-free Adaptation (ICML25)

[15] Efficient Vision Transformer via Token Merger (TIP23)

---

> ### Author Response · Authors · 2025-12-06
> **Response**
>
> Thank you so much for the detailed and constructive comments. We address your concerns as follows.
>
> ---
>
> > **W1**: These is no discussion on long-context compression methods in LLMs. The text modality is also one important modality. There are many important context compression methods, which inspire other methods in the multi-modal area.
>
>
> **Response to W1**: We thank the reviewer for this insightful suggestion. While we acknowledge that text compression methods are important, we respectfully clarify that multimodal token compression differs significantly from pure text approaches. This distinction arises because multimodal compression is intrinsically tied to modality encoders and cross-modal fusion. Given that comprehensive surveys on long-context text compression already exist (as acknowledged in our Introduction, Page 2), this paper is deliberately scoped to focus on the unique challenges and solutions within the multimodal domain.
> Furthermore, Section 2.4 emphasizes the criteria for including methods in the taxonomy, guiding the reader to the essence of the survey's content.
>
> ---
>
> > **W2**: Miss the discussion on methods using sparse attention. The attention-based compression methods are similar to some works in sparse attention. A relevant discussion can make this survey better.
>
>
> **Response to W2**: We fully agree that attention-based compression methods resonate with the principles of sparse attention. We have updated the manuscript (in page 10) to include a discussion on these relevant works and enriched the survey for a broader audience.
>
> > **W3**: The discussion on linear attention is also important for multi-modal context compression. Especially, the recent Qwen3-Next has proven its potential and ability.
>
> ---
>
> **Response to W3**: We fully agree with your insight. While our discussed token compression method effectively addresses the computational bottleneck by mitigating the $O(N^2)$ complexity of full attention, we recognize that linear attention represents a promising parallel direction for achieving efficiency. We have incorporated a discussion of this topic in the revised manuscript (in page 24) to provide readers with a more comprehensive perspective on the efficient AI landscape.
>
> ---
>
> > **Requested Changes**
>
> We are very willing to include these related articles in our survey to help readers better understand the relevant domain. We appreciate the reviewer's suggestion.

---

### Review · Reviewer_qXfH · 2025-11-27

**Summary Of Contributions:**

This paper provides a systematic survey on Multimodal Long-Context Token Compression, an emerging and critical area aiming to address the computational bottlenecks (specifically the quadratic complexity of self-attention) in MLLMs. The survey covers a wide range of recent methodologies, effectively bridging the gap between traditional token compression in LLMs/ViTs and the specific needs of MLLMs.

**Audience:**

Yes

**Audience Explanation:**

Multimodal LLMs are currently a dominant topic in the ML community. The specific focus on "long-context" and "efficiency" targets one of the most significant bottlenecks in deploying these models today. The paper covers vision, audio, and language, making it relevant to a broad audience across these subfields.

**Broader Impact Concerns:**

No major ethical concerns.

**Claims And Evidence:**

Yes

**Claims Explanation:**

Strengths:

- With the rapid growth of long-context MLLMs, this survey addresses a very urgent need for efficiency.
- The categorization is intuitive. The distinction between modality-specific constraints (e.g., temporal redundancy in video vs. spatial in images) and algorithmic principles is well-articulated.
- The paper includes very recent works, making it a current and relevant resource.

Weaknesses:

- The survey functions more as an enumerated bibliography than a critical synthesis; it fails to apply the analytical framework proposed in Table 1 (Pros & Cons) to the subsequent discussions of specific methods.
- The proposed taxonomy fundamentally conflates two distinct research paradigms—architecturally embedded compression (requiring training) and post-hoc inference optimization (training-free)—which obscures practical distinctions for the reader.
- The discussion on evaluation methodologies is insufficient, lacking a critical assessment of whether current benchmarks (e.g., VQA) are sensitive enough to measure the degradation caused by aggressive compression.

**Requested Changes:**

I suggest the following revisions to elevate the manuscript from a descriptive list to a high-quality analytical survey.

First, I suggest that the authors integrate the analytical insights established in Table 1 into the core narrative of Sections 3, 4, and 5. Currently, Table 1 provides a strong theoretical framework regarding the pros and cons of different mechanisms, but this framework is largely utilized only in the introduction. It would be beneficial to refine the descriptions of specific methods to explicitly critique them using these criteria. For instance, when discussing a transformation-based method, the text could analyze how it might suffer from the "inflexible compression rate" limitation noted in Table 1, thereby providing the synthesis expected of a comprehensive survey.

Second, I recommend expanding Section 5 to rectify the current imbalance in modality coverage. The analysis of audio-centric compression is currently disproportionately brief compared to the visual sections. The survey would be significantly strengthened by increasing the depth of this section to match Section 3. This could include adding a dedicated subsection discussing the unique types of redundancy specific to audio—such as spectral versus temporal redundancy, silence, or repetitive noise—and analyzing how current methods specifically address these audio-specific challenges.

Third, I suggest providing a more rigorous and critical discussion regarding evaluation protocols. Simply listing benchmarks in Table 4 offers limited insight. It would be valuable to add a critical analysis of the "evaluation gap," specifically addressing why general-purpose benchmarks like VQA or ActivityNet may be inadequate for evaluating compression performance, as they often require only low-fidelity information. I encourage the authors to advocate for, or at least discuss, the need for high-fidelity metrics (such as OCR or fine-grained temporal grounding) that are sensitive enough to detect the loss of critical details caused by token compression.

Finally, regarding Section 6.3, the claim that combining compression methods is generally ineffective appears to be currently unsupported by sufficient evidence. I recommend that the authors either provide concrete empirical evidence to support this broad assertion or refine the argument to differentiate between overlapping strategies (which may conflict) and orthogonal strategies (which theoretically should synergize). Referencing recent works that explicitly explore such combinations would provide a more nuanced and complete perspective.

---

> ### Author Response · Authors · 2025-12-06
> **Response**
>
> Thank you so much for the detailed and constructive comments. We address your concerns as follows.
>
> ---
>
> > **W1 and R1**: The survey functions more as an enumerated bibliography than a critical synthesis; it fails to apply the analytical framework proposed in Table 1 (Pros & Cons) to the subsequent discussions of specific methods.
>
> **Response to W1 and R1**: (The W1 and R1 are similar. We respond to them collectively here.) Our original rationale was that, given the structural similarity (i.e., modality encoders and LLM decoders) across image, video, and audio, the four method categories share the same pros and cons, so we focused the core analysis and discussion *primarily in Section 3 (Image Modality)* as a foundation (specifically in Sections 3.1.4, 3.2.1, 3.3.3, and 3.4.3).
>
> To better address your concern and deliver the expected synthesis, we have carefully revised the manuscript to strengthen the critical synthesis, ensuring that the discussions of specific methods are tightly linked to the pros and cons outlined in Table 1. These revisions optimize the descriptive flow and are highlighted in *olive* in the revised manuscript for your convenience.
>
> ---
>
> > **W2**: The proposed taxonomy fundamentally conflates two distinct research paradigms—architecturally embedded compression (requiring training) and post-hoc inference optimization (training-free)—which obscures practical distinctions for the reader.
>
> **Response to W2**: We appreciate the reviewer for pointing out this crucial distinction. While we agree that the operational requirements (training-based vs. training-free) differ, we respectfully argue that the boundary between these paradigms is becoming fluid in recent research.
>
> Many methods characterized as "training-free" are not intrinsically limited to training-free optimization; rather, they serve as initialization or architectural baselines that can—and often do—incorporate fine-tuning to enhance modality alignment and performance. For instance:
>
> - VisionZip[1] proposes a training-free token reduction but achieves its state-of-the-art performance (VisionZip$^\ddagger$) by fine-tuning the projector to realign compressed visual tokens with the LLM space.
> - FastVID[2] serves as a training-free adaptation of Chat-UniVi[3], yet its principles apply to trained settings.
> - VideoChat-Flash[4] validates various pruning strategies that are initially training-free but verified through training.
>
> Therefore, to avoid artificially separating methods that share underlying mechanisms, we structured the taxonomy based on algorithmic logic rather than training status. However, to address your concern regarding practical clarity for the reader, we have explicitly labeled each paper as "Training-free" or "Training-based" in our maintained GitHub repository to provide a clear user guide.
>
> ---
>
> > **W3 and R3**: The discussion on evaluation methodologies is insufficient, lacking a critical assessment of whether current benchmarks (e.g., VQA) are sensitive enough to measure the degradation caused by aggressive compression.
>
> **Response to W3 and R3**: Thank you for pointing out this. We have updated Section 6.4.4 to critically discuss the sensitivity of current benchmarks and have supplemented our analysis with more suitable evaluation schemes to quantify degradation accurately.
>
> ---
>
> > **R2**: I recommend expanding Section 5 to rectify the current imbalance in modality coverage. The analysis of audio-centric compression is currently disproportionately brief compared to the visual sections. The survey would be significantly strengthened by increasing the depth of this section to match Section 3. This could include adding a dedicated subsection discussing the unique types of redundancy specific to audio—such as spectral versus temporal redundancy, silence, or repetitive noise—and analyzing how current methods specifically address these audio-specific challenges.
>
> **Response to R2**: We sincerely thank the reviewer for this constructive suggestion. We have expanded Section 5 to rectify the imbalance in modality coverage. To our best knowledge that the volume of existing literature on audio token compression is less than that for visual modalities, but we agree that a deeper analysis is necessary.
>
> Accordingly, we have added a dedicated Subsection 5.5, which discusses audio-specific redundancies (e.g., spectral vs. temporal redundancy, silence, and repetitive noise) and analyzes the unique challenges in this domain. We have also incorporated recent relevant studies to strengthen the comprehensive nature of this section.
>
> ---

---

> > ### Author Response · Authors · 2025-12-06
> > **Response (Continued)**
> >
> > > **R4**: Regarding Section 6.3, the claim that combining compression methods is generally ineffective appears to be currently unsupported by sufficient evidence. I recommend that the authors either provide concrete empirical evidence to support this broad assertion or refine the argument to differentiate between overlapping strategies (which may conflict) and orthogonal strategies (which theoretically should synergize). Referencing recent works that explicitly explore such combinations would provide a more nuanced and complete perspective.
> >
> > **Response to R4**: We thank the reviewer for this constructive suggestion. In our revised manuscript, we have rewritten Section 6.3 to provide a more comprehensive discussion. Specifically, we have categorized the combinations of multiple methods into three distinct types: orthogonal, conflicting, and those requiring co-design. We believe this classification better illustrates the potential and constraints of integrating different approaches.
> >
> > [1] Visionzip: Longer is better but not necessary in vision language models (CVPR 2025)
> >
> > [2] FastVID: Dynamic Density Pruning for Fast Video Large Language Models (NeurIPS 2025)
> >
> > [3] Chat-UniVi: Unified Visual Representation Empowers Large Language Models with Image and Video Understanding (CVPR 2024 Highlight)
> >
> > [4] VideoChat-Flash: Hierarchical Compression for Long-Context Video Modeling (arXiv 2501.00574)

---

> ### Comment · Action_Editor_yaH7 · 2025-12-21
>
> Dear reviewer,
>
> Can you input your final recommendation?
>
> Best, AE

---

### Decision · Action_Editor_yaH7 · 2025-12-25

**Recommendation:** Accept as is

**Audience:**

Yes

**Audience Explanation:**

Multimodal LLMs are currently a dominant topic in the ML community. The specific focus on "long-context" and "efficiency" targets one of the most significant bottlenecks in deploying these models today. The paper covers vision, audio, and language, making it relevant to a broad audience across these subfields.

**Claims And Evidence:**

Yes

**Claims Explanation:**

This paper presents the a systematic survey on multimodal long-context token compression, addressing the computational bottlenecks in MLLMs caused by the long context generated from high-resolution images, long videos, and audio. The authors propose a structured taxonomy that categorizes existing methods first by modality (image, video, and audio) to address specific redundancies, and further by their underlying mechanisms, such as transformation-based, similarity-based, attention-based, and query-based approaches.

During the rebuttal, the authors actively responded to the reviewers' comments and addressed their concerns:

- Addressing concerns regarding the lack of critical synthesis and modality imbalance (especially in audio);

- Incorporating the quantitative trade-off analysis (Figure 4);
- Adding discussions on sparse and linear attention.

All three reviewers are satisfied with the manuscript and recommended acceptance of the paper.